# Characterisation of low-base and mid-base clouds and their thermodynamic phase over the Southern Ocean and Arctic marine regions

Barbara Dietel[1], Odran Sourdeval[2], and Corinna Hoose[1]

[1]Institute of Meteorology and Climate Research Troposphere Research, Karlsruhe Institute of Technology, Karlsruhe, Germany

[2]University of Lille, CNRS, UMR 8518-LOA-Laboratoire d'Optique Atmosphérique, F-59000 Lille, France

**Correspondence:** Barbara Dietel (barbara.dietel@alumni.kit.edu), Corinna Hoose (corinna.hoose@kit.edu)

**Abstract.** The thermodynamic phase of clouds in low and middle levels over the Southern Ocean and the Arctic marine regions is poorly known, leading to uncertainties in the radiation budget in weather and climate models. To improve the knowledge of the cloud phase, we analyse two years of the raDAR-liDAR (DARDAR) dataset based on active satellite instruments. We classify clouds according to their base and top height and focus on low-, mid- and mid-low-level clouds as they are most frequent in the mixed-phase temperature regime. Low-level single-layer clouds occur in 8-15 % of all profiles, but single-layer clouds spanning the mid-level also amount to approx. 15 %. Liquid clouds show mainly a smaller vertical extent, but a horizontally larger extent compared to ice clouds. The results show the highest liquid fractions for low-level and mid-level clouds. Two local minima in the liquid fraction are observed around cloud top temperatures of -15 °C and -5 °C. Mid-level and mid-low-level clouds over the Southern Ocean and low-level clouds in both polar regions show higher liquid fractions if they occur over sea ice compared to open ocean. Low-level clouds and mid-low-level clouds with high sea salt concentrations, used as a proxy for sea spray, show reduced liquid fractions. In mid-level clouds, dust shows the largest correlations with liquid fraction with a lower liquid fraction for a higher dust aerosol concentration. Low-level clouds clearly show the largest contribution to the shortwave cloud radiative effect in both polar regions followed by mid-low-level clouds.

## 1 Introduction

Cloud phase has a major influence on the Earth's radiation budget due to different scattering, absorption and emission properties of liquid droplets and ice particles. Weather and climate models show strong biases in the shortwave radiation over the Southern Ocean related to biases in the representation of clouds and their phase (e.g. Cesana et al., 2022; Forbes and Ahlgrimm, 2014). Although in the recent sixth phase of the Coupled Model Intercomparison Project (CMIP6), models have improved their representation of clouds compared to the results of CMIP5, the cloud feedback remains the largest source of uncertainty in climate feedbacks (Arias et al., 2021; Forster et al., 2021). The CMIP5 results (Taylor et al., 2012) showed large radiative biases over the Southern Ocean caused by a lack of supercooled liquid water mainly in the cold sector of cyclones (Arias et al., 2021; McFarquhar et al., 2021; Bodas-Salcedo et al., 2016). Cesana et al. (2022) showed that CMIP6 simulations reduced the

average radiative bias over the Southern Ocean by increasing the number of low- and mid-level clouds. Models with more complex microphysics than only temperature dependent liquid and ice partitioning tend to show a better representation of the liquid phase fraction, but all models struggle to generate the correct shortwave reflection south of $55\,°S$ (Cesana et al., 2022). Zelinka et al. (2020) showed that the spread of radiative cloud feedback between different models increased from CMIP5 to CMIP6. Contrary to previous results, where "too few and too bright" stratocumulus clouds were simulated, CMIP6 results show too many stratocumulus clouds over the Southern Ocean, which are not bright enough compared to observations (Schuddeboom and McDonald, 2021). To improve the radiative balance over the Southern Ocean, the accurate representation of stratocumulus clouds in simulations should remain a priority (Schuddeboom and McDonald, 2021). Desai et al. (2023) showed the underestimation of the ice phase below cloud top in low-level clouds over the Southern Ocean in the Energy Exascale Earth System Model version 1 (E3SMv1). Besides low-level clouds, CMIP6 models also struggle to simulate the correct optical properties and regime variability of mid-level topped clouds forming in the boundary layer over the Southern Ocean leading to errors in the shortwave reflection (Cesana et al., 2022). Based on the different definitions of cloud types, the results of previous papers differ in finding the source of the radiative biases either in low-level stratocumuls clouds only (Schuddeboom and McDonald, 2021) or in low-level and mid-level clouds (Cesana et al., 2022). Furthermore, there is a lack of studies investigating mid-level clouds, which were described as the "forgotten clouds" by Vonder Haar et al. (1997). Until now the number of studies on mid-level clouds (Alexander and Protat, 2018; Kayetha and Collins, 2016; Mason et al., 2014; Sassen and Wang, 2012; Zhang et al., 2010; Smith et al., 2009; Fleishauer et al., 2002) is limited and they often only examine case studies, or focus on specific mid-level clouds, such as optically thin ice clouds or liquid-layer topped clouds.

Besides the large errors over the Southern Ocean, the representation of low-level clouds in the Arctic also shows large uncertainties in model simulations (Taylor et al., 2019). Taylor et al. (2019) also pinpoint the need of an improved understanding of the ice formation and the cloud phase partitioning. Wei et al. (2021) showed a too high cloud fraction over the Arctic in CMIP6 simulations compared to various satellite observations leading to the underestimation of the shortwave radiation at the surface. Tjernström et al. (2021) showed that the Integrated Forecasting System (IFS) of the European Centre for Medium-Range Weather Forecasts (ECMWF) shows too much cloud occurrence below 3km and not enough between 3km and 5km. A too high cloud cover below 3km was also seen by McCusker et al. (2023) in the Met Office Unified Model (UM) and in the IFS. Comparing model results with observations from the Multidisciplinary drifting Observatory for the Study of Arctic Climate expedition (MOSAiC, see Shupe et al. (2022)) during winter, all operational and experimental forecast system models underestimated the liquid water path (LWP) and overestimated the ice water path (IWP) (Solomon et al., 2023). Klein et al. (2009) showed as well too small LWP in cloud-resolving model and single-column model simulations compared to observations for a case study of Arctic single-layer clouds, but simulated IWP being generally consistent with observations.

Cloud phase is influenced by the temperature with increasing freezing probability for lower temperatures. Nevertheless, supercooled liquid clouds occur quite frequently in polar regions at lower temperatures (D'Alessandro et al., 2021; Bodas-Salcedo et al., 2016; Hu et al., 2010). Besides temperature, cloud phase is also influenced by the availability of ice nucleating particles (INPs) and their composition, as well as by other ice formation processes depending on liquid particle properties and supersaturation. Especially marine organics including biological parts seem to play an important role as INPs in polar regions

(Ickes et al., 2020; McCluskey et al., 2018; DeMott et al., 2016; Wilson et al., 2015; Burrows et al., 2013). A previous study suggested an influence of sea ice coverage on the availability of sea spray aerosols and thereby an effect on the cloud phase of low-level clouds over both hemispheres (Carlsen and David, 2022). Papakonstantinou-Presvelou et al. (2022) found higher ice number concentrations in Arctic low-level ice clouds over sea ice compared to Arctic low-level clouds over the ocean. Zhang et al. (2019) and Lenaerts et al. (2017) compare clouds between both polar regions. Zhang et al. (2019) use ground-based remote sensing instruments while Lenaerts et al. (2017) compare satellite observations with reanalysis and climate simulations.

Due to the lack of a consistent cloud classification, it is hard to compare the results of studies using different classifications. Cloud classifications are often dependent on the datasets used in the studies. Passive satellite observations mainly provide cloud top information or vertically integrated cloud water contents, while active instruments provide vertically resolved information, but have a less good temporal resolution compared to geostationary passive satellite observations. Many studies based on passive satellite observations are based on the ISCCP cloud regimes (Rossow and Schiffer, 1999) only using cloud top pressure and optical thickness for the cloud classifications. Even for studies using active instruments, cloud type definitions vary, as e.g. low-level clouds are defined by the cloud top height (CTH) smaller or equal than $3\,\mathrm{km}$ in Danker et al. (2022), while many CloudSat products use the classification of Wang and Sassen (2001), which includes various parameters like the cloud base height (CBH), rain, horizontal and vertical cloud dimensions, and the LWP. There is certainly a need for a more uniform definition of specific cloud types and the parameters they depend on. This study investigates all cloud types having a low or middle cloud base height, but also considers various cloud tops enabling a more detailed analysis considering various cloud extensions. To distinguish cloud types, we use the cloud top height as well as cloud base height and divide the troposphere into 3 layers (low, middle, high) similar to the definitions by the World Meteorological Organization (2017). Detailed definitions are described in Sec. 3.

The main objective of this paper is to systematically investigate the occurrence of cloud types with low and middle cloud base heights and to improve the knowledge about their phase distribution over the Southern Ocean and the Arctic marine regions. Section 2 describes the data which are used for this analysis. Section 3 follows with a description of the methods used to distinguish cloud types and to investigate the cloud phase. The main results are then described and discussed in Sec. 5, split into different subsections. Section 5.1 presents the frequency of occurrence of the different cloud types and Sec. 5.2 focuses on the thermodynamic phase of the different cloud types with several subsections investigating the influence and correlation of various parameters on the cloud phase. Section 5.3 covers the cloud radiative effects of the different cloud types and its correlation with the cloud phase. Section 4 states some uncertainties of the datasets, the method, and the results. Lastly, Sec. 6 summarises the major results.

## 2 Data

### 2.1 Cloud phase classification

The raDAR-liDAR (DARDAR) dataset is based on measurements of the CloudSat (Stephens et al., 2002) satellite based radar and the Cloud-Aerosol Lidar with Orthogonal Polarization (CALIOP) onboard the Cloud-Aerosol Lidar and Infrared Pathfinder

Satellite Observation (CALIPSO) satellite (Winker et al., 2009). DARDAR provides a detailed cloud phase categorisation (see Tab.1) based on a retrieval scheme using a combination of the radar reflectivity and the lidar backscatter. While the radar is more sensitive to ice particles due to their larger size, the lidar shows a strong enhancement in the backscatter when liquid droplets are observed. Details on the retrieval of the phase categorisation can be found in Delanoë and Hogan (2010, 2008). For our study we use the second version of the DARDAR classification described and validated in Ceccaldi et al. (2013). Furthermore, collocated ECMWF AUXillary (ECMWF-AUFX) data are used within the retrieval process to categorise the cloud phase (Delanoë and Hogan, 2010). Temperature from this dataset is included in the DARDAR dataset and also used for further analysis. The dataset has a vertical resolution of $60\,\mathrm{m}$ and a horizontal resolution of about $1.5\,\mathrm{km}$ along the track of the polar-orbiting satellites.

## 2.2 Sea ice data

We use daily sea ice concentrations from Nimbus-7 SMMR and DMSP SSM/I-SSMIS Passive Microwave Data, Version 1 (Cavalieri et al., 1996), which have an original horizontal resolution of $25\,\mathrm{km}$. This is much coarser compared to the DARDAR dataset, but is suitable for our analysis focusing on the influence of aerosol emissions distributed in the boundary layer. As sea ice is also temporally more homogeneous than clouds, we think a daily resolution of sea ice is sufficient for our purpose. Furthermore, while the coarse resolution is not sufficient to identify small fractures in sea ice coverage, it can provide general information on the coverage of the ocean by sea ice. The advantage of the microwave instrument is that the sea ice concentrations can also be observed in cases where clouds are above the sea ice, which is of special interest for our study investigating the correlation of sea ice with the cloud phase.

## 2.3 Aerosol data

The ECMWF Atmospheric Composition Reanalysis 4 (EAC4) (Inness et al., 2019) from the Copernicus Atmosphere Monitoring Service (CAMS) provides mixing ratios of different aerosol types: dust, sea salt, organic matter, black carbon, and sulphate aerosol. The horizontal resolution is about $80\,\mathrm{km}$ and the temporal resolution is 3-hourly (Inness et al., 2019). The aerosol optical depth (AOD) from the Moderate Resolution Imaging Spectroradiometer (MODIS) and from the Advanced Along-Track Scanning Radiometer (AATSR) is assimilated by a 4D-Var data assimilation system of ECMWF's IFS (Inness et al., 2019). Lapere et al. (2023) compare CAMS sea salt reanalysis with a few station observations in the Arctic and Antarctic and show in their Fig. 7 that most stations show strong Pearson correlation coefficients despite partly high normalised mean biases.

## 2.4 Cloud radiative effects

We analyse the cloud radiative effect (CRE) of various cloud types at the top of the atmosphere (TOA) using the version P1_R05 of the CloudSat 2B-FLXHR-LIDAR product (Henderson et al., 2013; L'Ecuyer et al., 2008). The cloud radiative effect is defined as the difference between the net flux in clear-sky conditions and the net flux in all-sky conditions. The net flux is calculated as the difference between the upward flux ($F_\uparrow$) and the downward flux ($F_\downarrow$) (see Eq. 1) (L'Ecuyer et al.,

2008).

$$CRE = (F_\uparrow - F_\downarrow)_{\mathrm{clear-sky}} - (F_\uparrow - F_\downarrow)_{\mathrm{all-sky}} \tag{1}$$

The radiative transfer algorithm estimates radiative fluxes based on CloudSat, CALIPSO, and MODIS products including cloud information like the ice water content, the liquid water content and the effective radius, precipitation information, but also aerosol information from CALIPSO, as well as temperature and humidity profiles from ECMWF analyses (Henderson et al., 2013). Further flux calculations are performed after removing all clouds to calculate the CRE.

The net cloud radiative effect (NETCRE) is calculated by summing up the longwave cloud radiative effect (LWCRE) and the shortwave cloud radiative effect (SWCRE).

## 2.5 Time and region

We use data from all products between the years 2007 and 2008, because the data availability of CloudSat/CALIPSO products is optimal for these years as there are no large data gaps due to instrument failures or satellite issues. The region of the Southern Ocean is defined as latitudes from $40\,°\mathrm{S}$ to $82\,°\mathrm{S}$ similar to previous studies (Cesana et al., 2023; D'Alessandro et al., 2021; Bodas-Salcedo et al., 2016). The poleward boundary is based on the availability of the dataset from active sensors on polar orbiting satellites. Data over land are excluded, which leads to a boundary between $70\,°\mathrm{S}$ and $80\,°\mathrm{S}$, following the coast of Antarctica. The Arctic marine regions are defined from $60\,°\mathrm{N}$ to the northern boundary of the dataset at around $82\,°\mathrm{N}$, similar to the definition of the Arctic Ocean by the International Hydrographic Organization (Jakobsson et al., 2004). Both regions cover various conditions in the parameter space of different surface conditions, aerosol contents, and meteorological dynamical conditions. Both regions are illustrated in Fig. 1 highlighting the sea ice variation between the seasons.

## 3 Methodology

As a first step, we collocate the DARDAR data with the 2B-CLDCLASS-LIDAR dataset, which means that only profiles are considered that occur in both datasets. We only consider profiles with a single cloud layer to reduce uncertainties and the influence of overlapping clouds. This means profiles that contain a single connected cloud layer from the phase mask of DARDAR without any gaps in between, as discontinuities don't allow a continuous analysis of e.g. the vertical phase distribution within a cloud. Thereby, multi-layer clouds are excluded from this study, which reduces the number of cloud profiles by about $50\,\%$. Table 1 shows all categories from the DARDAR-MASK v2 classification and describes which of them we consider as a cloud and as which thermodynamic cloud phase they are assigned to. We calculate the CBH, the CTH, and the vertical extent for each cloud profile.

To distinguish between cloud types, we use the CBH and the CTH. Figure 2 shows a schematic of the cloud types we investigate in this paper. The troposphere is divided into three layers. The lower troposphere layer is defined from $0\,\mathrm{km}$ to $2\,\mathrm{km}$. The middle troposphere layer is defined from $2\,\mathrm{km}$ to a threshold $Z_{\mathrm{max}}$, which increases from $4\,\mathrm{km}$ at the pole to $7\,\mathrm{km}$ at $40\,°$ based on the latitude ($lat$), see Eq. (2). The highest layer in the troposphere is then consequently defined by heights

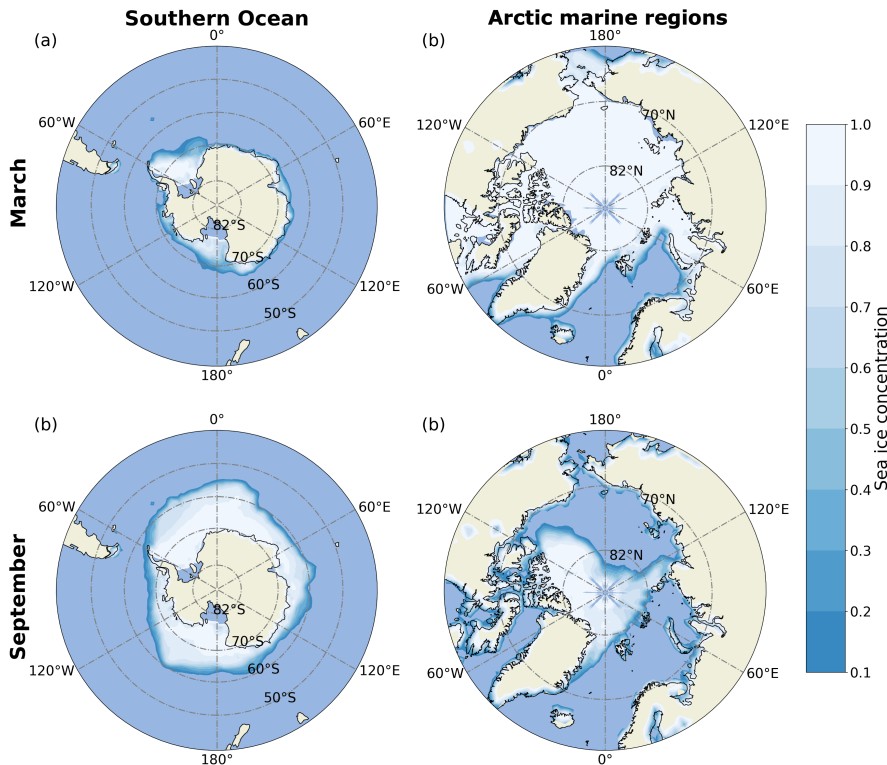

**Figure 1.** Map of the Southern Ocean and the Arctic marine regions with sea ice concentrations in March and September averaged for 2007 and 2008 to illustrate sea ice cover over both regions. Data are from (Cavalieri et al., 1996) (version 1) provided by the National Snow and Ice Data Centre (NSIDC).

| Lowlevel | Mid-lowlevel | High-mid-lowlevel | Midlevel | High-midlevel |
|---|---|---|---|---|

Z$_{max}$

2 km

0.5 km

**Figure 2.** Schematic of the different cloud types classified by their top and base heights. The upper threshold $Z_{max}$ is a value between $4\,km$ and $7\,km$ depending on the latitude and described in Sec. 3.

**Table 1.** DARDAR-MASK categories and how they are considered for the cloud phase analysis in this study

| DARDAR-MASK v2 | Considered Phase |
|---|:---:|
| Presence of liquid unknown | - |
| Surface and subsurface | - |
| Clear sky | - |
| Ice clouds | Ice |
| Spherical or 2D ice | Ice |
| Supercooled water | Liquid |
| Supercooled + ice | Mixed |
| Cold rain | - |
| Aerosol | - |
| Warm rain | - |
| Stratospheric clouds | - |
| Highly concentrated ice | Ice |
| Top of convective towers | - |
| Liquid cloud | Liquid |
| Warm rain + liquid clouds | Liquid |
| Cold rain + liquid clouds | Liquid |
| Rain may be mixed with liquid | Liquid |
| Multiple scattering due to supercooled water | Liquid |

larger than the previously described threshold $Z_{\max}$. The two thresholds $Z_{\max}$ and $2\,\mathrm{km}$ defining the three tropospheric layers, are also shown in Fig. 3 with the dashed lines. The thresholds for the definitions are based on the definitions from the World

Meteorological Organization (2017). Regarding the vertical distribution of the annual mean temperatures in Fig. 3, the threshold $Z_{\max}$ is also mostly parallel to the isotherms, which shows one of the reasons for the chosen threshold decreasing polewards. Furthermore, the threshold $Z_{\max}$ is in the upper part of the mixed-phase temperature regime.

$$Z_{\max} = \begin{cases} 4\,\mathrm{km} + \frac{7\,\mathrm{km} - 4\,\mathrm{km}}{-40\,° - (-90\,°)} \cdot (90\,° - lat) & \text{,if } lat > 0. \\ 4\,\mathrm{km} + \frac{7\,\mathrm{km} - 4\,\mathrm{km}}{-40\,° - (-90\,°)} \cdot (90\,° + lat) & \text{,if } lat < 0. \end{cases} \tag{2}$$

Low-level clouds (L) are defined by a CBH and CTH between $500\,\mathrm{m}$ and $2\,\mathrm{km}$, compare first column in Fig. 2. Clouds with

160 CBH below $500\,\mathrm{m}$ are excluded from the study to reduce uncertainties introduced by ground clutter in the CloudSat signal (Bertrand et al., 2024; Liu, 2022; Blanchard et al., 2014). The second column in Fig. 2 shows mid-low-level clouds (ML) with CBH between $500\,\mathrm{m}$ and $2\,\mathrm{km}$ and CTH between $2\,\mathrm{km}$ and $Z_{\max}$. Clouds with CBH between $500\,\mathrm{m}$ and $2\,\mathrm{km}$ and CTH larger than $Z_{\max}$ are called high-mid-low-level clouds (HML), see third column in Fig. 2. Mid-level clouds (M) have their

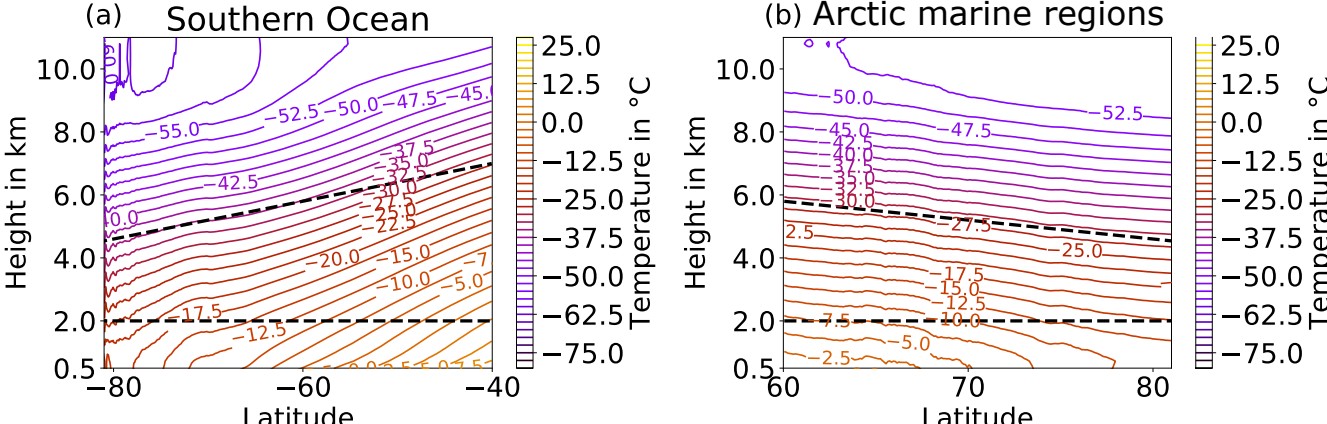

**Figure 3.** Zonal mean temperature over the Southern Ocean (a) and the Arctic marine regions (b). The coloured contours show the zonal mean temperature averaged between 2007 and 2008. The data are based on ECMWF reanalysis data, which are collocated to the DARDAR dataset. The black dashed lines indicate the thresholds of the atmosphere layers, to define the different cloud types. The lower threshold is at $2\,\mathrm{km}$, the upper threshold height decreases polewards following Eq. 2.

CBH and their CTH between $2\,\mathrm{km}$ and $Z_{\mathrm{max}}$, shown in the forth column in Fig. 2. The last column in Fig. 2 shows high-mid-level clouds (HM) with CTH larger than $Z_{\mathrm{max}}$ and CBH between $2\,\mathrm{km}$ and $Z_{\mathrm{max}}$. We generally use two approaches, (1) the statistical analysis of individual cloud profiles, and (2) the analysis of cloud objects, which are horizontally connected cloud profiles. In most parts of the paper approach (1) is used, while in Sec. 5.2.1 approach (2) is used.

The liquid fraction $f$ is calculated for each vertical cloud column by

$$f = \frac{n_{\mathrm{liq}} + 0.5 \cdot n_{\mathrm{mix}}}{n_{\mathrm{liq}} + n_{\mathrm{mix}} + n_{\mathrm{ice}}} \tag{3}$$

with $n_{\mathrm{liq}}$ being the number of liquid vertical bins of the cloud profile, $n_{\mathrm{mix}}$ being the number of mixed-phase vertical bins, and $n_{\mathrm{ice}}$ as the number of ice vertical bins. Due to the absence of better information it is assumed that half of each mixed-phase vertical bin consists of liquid droplets and half of it consists of frozen ice crystals, as the mixed-phase category is mainly based on a signal from both the radar and the lidar. Based on these calculations, a liquid fraction which is larger than 0 and less than 1 refers to mixed-phase cloud profiles. In the following, „liquid fractions" or „vertical liquid fractions" refer to this definition of $f$, unless otherwise stated.

The cloud top liquid fraction $f_{\mathrm{CT}}$ is calculated with Eq. 4 only considering the cloud top phase, i.e., the uppermost 60-m layer in each column that DARDAR defines as cloud. In Eq. 4 $N_{\mathrm{liq}}$ is the total number of liquid cloud top bins, $N_{\mathrm{mix}}$ is the total number of mixed-phase cloud top bins, and $N_{\mathrm{ice}}$ is the total number of ice bins at cloud top. $f_{\mathrm{CT}}$ is not meaningful for a single cloudy bin, but is statistically evaluated for all clouds with a given cloud top temperature.

$$f_{\mathrm{CT}} = \frac{N_{\mathrm{liq}} + 0.5 \cdot N_{\mathrm{mix}}}{N_{\mathrm{liq}} + N_{\mathrm{mix}} + N_{\mathrm{ice}}} \tag{4}$$

We use the sea ice concentration closest to each cloud profile to analyse a possible influence of sea ice on the cloud phase (see Sec. 5.2.4). We further distinguish between open and sea-ice-covered ocean. Open ocean is defined by a sea ice concentration of 0 %. For our analysis we consider sea ice concentration larger or equal than 80 % as sea-ice covered, which is based on the nomenclature from the World Meteorological Organisation (JCOMM Expert Team on Sea Ice, 2009). Over the Southern Ocean 75 % to 86 % of the cloud profiles occur over open ocean, 8 % to 14 % of the cloud profiles are over closed sea ice with the interval referring to the fraction for different cloud types. 6 % to 12 % of the cloud profiles over the Southern Ocean are over open ice with a sea ice concentration larger than 0 and lower than 80 %, which are not analysed in this study. Over the Arctic marine regions the fraction of cloud profiles over open ocean is much lower with 34 % to 37 %, a higher fraction of the cloud profiles (42 % - 49 %) is over sea ice, and 17 % -23 % of the cloud profiles occur over open ice. To investigate the differences in cloud phase depending on the sea ice concentration, the mean liquid fraction of cloud profiles over the open ocean are compared with the mean liquid fraction of cloud profiles over sea ice. For each CTT-bin of $2\,°C$ the distribution of the liquid fraction over ocean are compared to the distribution of the liquid fraction over sea ice, and it has been tested if the two distributions differ significantly using a Z-test and a p-value of 0.05.

We collocate the CAMS reanalysis to the DARDAR profiles. The mixing ratio of the different size modes of each aerosol type are summed up, to only consider aerosol types, without a size dependence. For the analysis the mean aerosol mixing ratios in the cloud profiles observed by the DARDAR dataset are calculated. We calculate the difference between the mean liquid fraction of clouds with a high aerosol type mixing ratio, greater than the 75th percentile, and the mean liquid fraction of clouds with a low aerosol type mixing ratio, less than the 25th percentile, as a function of the CTT with a bin size of $2\,°C$.

For both the ocean-sea ice comparison as well as the low-high aerosol comparison, we only calculate the difference of the liquid fractions if there are at least 500 cloud profiles in both categories for each CTT bin. These analysis are based on the statistics of individual profiles and no spatial analysis is done for this part.

In Sec. 5.2.1, we investigate the horizontal extent of the various cloud types and the correlation of the cloud phase with the horizontal extent. For this analysis cloud objects (approach (2)) are analysed. To examine the horizontal cloud extent, we analyse the time difference between a specific cloud profile and the next cloud profile of the same cloud type along the satellite track. If this time difference is less or equal than $0.2\,s$, which is the usual time difference to the next profile and corresponds to the resolution of the CloudSat profiles, the cloud profiles are considered as the same cloud. Time differences larger than $0.2\,s$ indicate a gap of about $1\,km$ between the cloud profiles, and the cloud profiles are therefore considered as separate clouds in absence of better information. Based on this separation, the phase of one cloud object, consisting of horizontally connected cloud profiles, is investigated. Only if all profiles are fully liquid, the cloud is considered as a liquid cloud. The same is valid for an ice cloud. Mixed-phase cloud objects either consist of ice profiles next to liquid profiles, or contain any mixed-phase profiles. In addition to the horizontal extent of the cloud, we also calculate the vertical extent by subtracting CBH from CTH for each vertical profile. For each cloud object, based on horizontally "connected" cloud profiles, we calculate the mean of the vertical extent of the single profiles to obtain the mean vertical extent of the cloud object. The horizontally connected profiles are only used for the analysis in Sec. 5.2.1. All other results are based on the single cloud profiles, no matter if they are connected or not.

## 4 Uncertainties

The combination of the different resolutions of the various datasets used in this study introduces uncertainties. While the DARDAR dataset has a very high spatial resolution ($\triangle h = 60\,\mathrm{m}, \triangle x = 1.5\,\mathrm{km}$), sea ice data and CAMS reanalysis have a much coarser spatial resolution. The temporal resolutions are also different, as the sea ice concentration is only available on a daily basis. Nevertheless, we think that due to the coarser resolution the temporal difference might play a minor role, as the sea ice concentration is only used to distinguish between sea ice conditions and open ocean. We don't consider small leads, cracks, etc., but rather want to distinguish between mostly sea-ice-covered ocean and open ocean.

For the aerosol reanalysis, the temporal resolution is 3-hourly and thereby introduces a temporal shift to the detection of single cloud profiles from the active satellite observations. Similar to the sea ice, the spatial resolution is rather coarse with $80\,\mathrm{km}$. However, we don't investigate the total value of a mixing ratio within single cloud profiles, but rather investigate them in a statistical sense considering the upper and lower concentration quartiles (see Sec.5.2.4). Furthermore, we investigate the mean of aerosol mixing ratios in specific cloud types.

One of the largest uncertainties regarding the phase detection is the strong attenuation and partly full extinction of the lidar signal in supercooled liquid layers. This uncertainty is well known, but in combination with studies using ground based observations the satellite perspective is still very useful and can lead to an improved understanding. Furthermore, we can interpret the liquid fraction used in this study as a lower boundary, which would even increase, if we could correct the lidar extinction and thereby the possible underestimation of the liquid phase. To examine the range of uncertainty, Fig. A1 and Fig. A2 show the results for the liquid fractions in the cloud types as a function of CTT, considering only profiles where the lidar is not fully attenuated, compared to considering all cloud profiles. Minor uncertainties are the extinction or attenuation of the radar signal in heavily raining clouds or deep convective systems, but as we focus more on shallow clouds and not tropical deep convective systems, this only plays a minor role. Another uncertainty is introduced by ground-clutter of the CloudSat radar signal. We have attempted to reduce the impact of ground clutter by excluding clouds with a cloud base below $500\,\mathrm{m}$, but also clouds above this threshold could be impacted to some extent (Blanchard et al., 2014; Marchand et al., 2008). Alexander and Protat (2018) showed that DARDAR underestimates clouds at heights of $0.2\,\mathrm{km}$ to $1.0\,\mathrm{km}$ by a factor of 3 compared to a surface-based lidar at Cape Grim, Australia from July 2013-February 2014.

An uncertainty in the analysis of the cloud radiative effect is based on the assumptions made in the retrieval process of the 2B-FLXHR-LIDAR dataset. As the retrieval assumes a certain profile including liquid water content and ice water content based on different CloudSat products, such as the 2B-CWC or 2B-GEOPROF, this may not be consistent with the DARDAR dataset, which we use to calculate the liquid fraction for each cloud profile. However, we expect only minor differences as both, the 2B-FLXHR-LIDAR retrieval and the DARDAR retrieval are mainly based on CloudSat and CALIPSO observations.

## 5 Results and discussion

### 5.1 Cloud type occurrence frequency

Figure 4 shows the occurrence frequency of the different cloud types in the two years 2007 and 2008 with respect to the total number of observed DARDAR profiles including cloud-free profiles. Low-level clouds are most frequent with an occurrence of 15.8 % over the Southern Ocean and 8.6 % over the Arctic marine regions, see magenta colour in Fig. 4. Mid-low-level clouds occur in about 6.5 % of the profiles over the Southern Ocean and in about 5.5 % over the Arctic marine regions. Mid-level clouds are much less frequent compared to low-level clouds. Nevertheless, they are not negligible and show similar frequencies as high-mid-level clouds, while high-mid-low-level clouds are slightly more frequent. The fraction of multi-layer and other clouds is slightly larger over the Southern Ocean with 44.7 % compared to 40.4 % over the Arctic marine regions. The "Multi-layer/Others" category is not comparable to a proper multi-layer cloud category, as there is neither a threshold for the minimum vertical extent of a cloud layer nor for the cloud-free layer. Even small cloud-free gaps of 60 m within a cloud layer fall into this category, because we exclude such profiles in our analysis as our focus in this study is on single-layer clouds and we want to reduce uncertainties introduced by discontinuities. Future studies could analyse the "Multi-layer / Others" category in more detail and investigate the sensitivity of the fractions to thresholds for a minimum vertical extent of cloud layers and gaps between cloud layers to distinguish between vertically overlapping cloud layers and smaller discontinuities in cloud layers. 12.6 to 15.1 % of the data are clouds with a CBH lower than 500 m which are not further analysed in this study because of large uncertainties due to ground clutter. Further analysis showed, that most of these cloud profiles have low cloud tops and would refer to the low-level cloud type category. The fraction of clear-sky profiles is higher over the Arctic marine regions with 19.9 % compared to 10.6 % over the Southern Ocean. Generally, the cloud type occurrence frequencies over the Southern Ocean are very similar to the ones over the Arctic marine regions except for a slightly higher occurrence of low-level and mid-low-level clouds over the Southern Ocean.

Sassen and Wang (2008) investigated the frequency of specific cloud types as a function of the latitude with the 2B-CLDCLASS dataset based on global CloudSat observations from 15 June 2006 to 15 June 2007 distinguishing between clouds over ocean and clouds over land. Although the cloud type definition and distinction criteria differ to the ones we use we still compare some similar cloud types. Our frequency of the low-level cloud category compares quite well to the stratus and stratocumulus (St+Sc) with frequencies between 20 and 30 % in similar latitudes over ocean. They find a higher fraction of high clouds between 1 % and 10 % compared to our result of about 1.4 %. The mid-level and mid-low-level clouds can be best compared to altocumulus (Ac) ($\approx 5\%$) and altostratus (As)($\approx 20\%$) clouds which both show higher frequencies compared to our results. The way how multi-layer clouds are considered in our study and the study of Sassen and Wang (2008) introduces a bias, because Sassen and Wang (2008) would still classify the overlapping cloud types, which leads to higher frequencies in their cloud categories compared to our results, as we consider them in a separate category as multi-layer clouds. This could explain some of the discrepancies between the observed numbers. Mace and Zhang (2014) showed a cloud coverage of about 35 % of clouds with cloud tops lower than 3 km over the Southern Ocean and 25 % over the Arctic marine regions, which is higher compared to our results, but the definitions differ to ours.

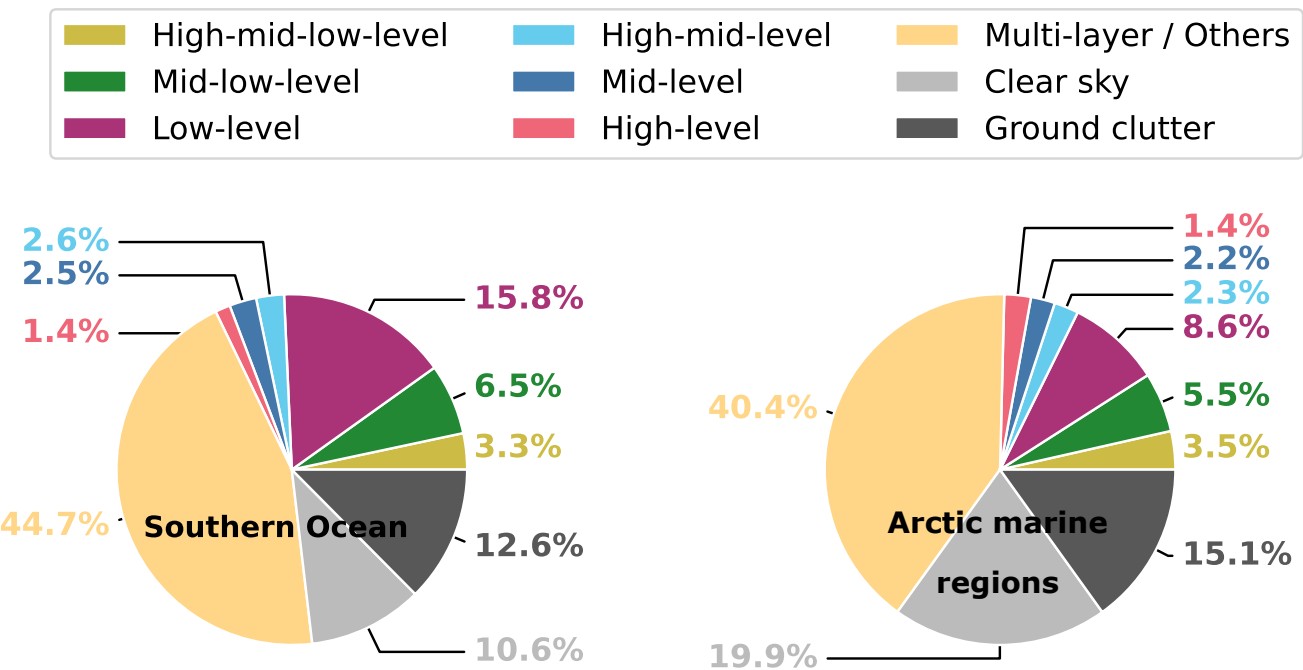

**Figure 4.** Frequency of different cloud type occurrence. The frequency is calculated as the total number of DARDAR-profiles containing a specific cloud type, observed during two years 2007 - 2008, and divided by the total number of DARDAR-profiles available including cloud-free profiles.

The temperature has a large influence on cloud phase, as mixed-phase clouds can only exist in the temperature range between 0 °C and -38 °C. Therefore, we investigate the relative frequencies of the different cloud types with respect to the total number of single-layer cloud profiles, as a function of their cloud top temperature (CTT), shown in Fig.5. In the mixed-phase temperature regime the fraction of low-level clouds is high. The relative frequency of low-level clouds is slightly higher over the Arctic marine regions compared to the Southern Ocean for CTT colder than -15 °C. Mid-low-level clouds also show a high frequency reaching a maximum of 0.6 over the Southern Ocean and 0.4 over the Arctic marine regions respectively at a temperature of around -30 °C. We see higher fractions of mid-level clouds over the Southern Ocean than over the Arctic marine regions for CTT colder than -15 °C, where already high-mid-level clouds exist over the Arctic marine regions. Regarding the mixed-phase temperature regime, low-level, mid-low-level and mid-level clouds are the dominating types of single-layer clouds in this temperature range. Therefore, further analysis and interpretations in this paper will focus on these cloud types.

## 5.2 Cloud phase

We will now investigate the thermodynamic phase of the different cloud types (Fig. 6).

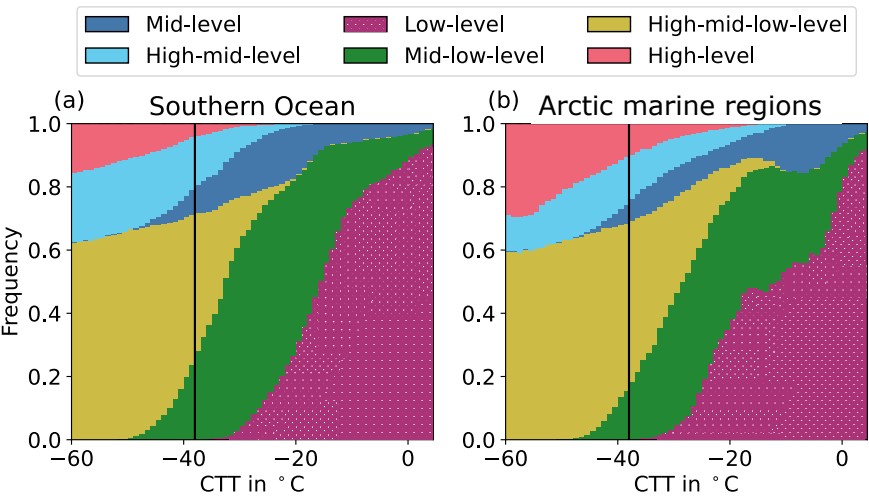

**Figure 5.** Stacked bars of the relative frequency of different cloud types only considering single-layer clouds as a function of cloud top temperatures (CTTs). The black line shows the lower temperature boundary of the mixed-phase temperature regime at -38 °C.

About half of low-level clouds are liquid clouds with a percentage of 40-53 %, half of them (41-52 %) are mixed-phase clouds, and a small fraction are pure ice clouds (6-9 %). The inner pie charts in Fig. 6 consider only cloud profiles where the lidar is not fully attenuated. We can see a decrease in the fraction of mixed-phase clouds to a percentage of 14-29 % for low-level clouds. Contrarily, the relative ice fraction and liquid fraction increases when considering only profiles where the lidar is not fully attenuated. Regarding the mid-low-level clouds a high fraction are mixed-phase profiles (72-74 %), a small fraction (6-10 %) are liquid profiles, while 18-20 % are ice profiles. High-mid-low-level are mainly ice profiles (60-74 %) and a smaller fraction (26-40 %) of mixed-phase profiles, but no liquid profiles. Mid-level clouds are liquid profiles in 36-37 % of the cases. A larger proportion of the mid-level clouds are mixed-phase profiles with a frequency of 37-41 %. 23-27 % of mid-level clouds are ice profiles. Regarding the differences for mid-level cloud profiles where the lidar signal is not fully attenuated the fraction of mixed-phase profiles decreases, while the fraction of ice profiles increases, and the fraction of liquid profiles slightly decreases over the Arctic marine regions. High-mid-level clouds show a similar distribution to high-mid-low-level clouds with large fraction of 64-78 % being ice profiles, and 21-35 % being mixed-phase profiles, but almost no liquid profiles (0-1 %) probably due to low cloud top temperatures with homogeneous ice formation.

Regarding the differences in the results considering all profiles of a cloud type, shown in the outer pie charts in Fig. 6 and only considering profiles where the lidar is not fully attenuated, we can generally see a decrease of the fraction of mixed-phase profiles, which indicates that in mixed-phase profiles the lidar is frequently attenuated. The reduction of the absolute numbers of profiles is strongest for mixed-phase clouds, which is the reason for the decrease of the relative fraction of mixed-phase profiles, while the relative fraction of ice and liquid profiles mostly increases. This might lead to uncertainties in the further analysis.

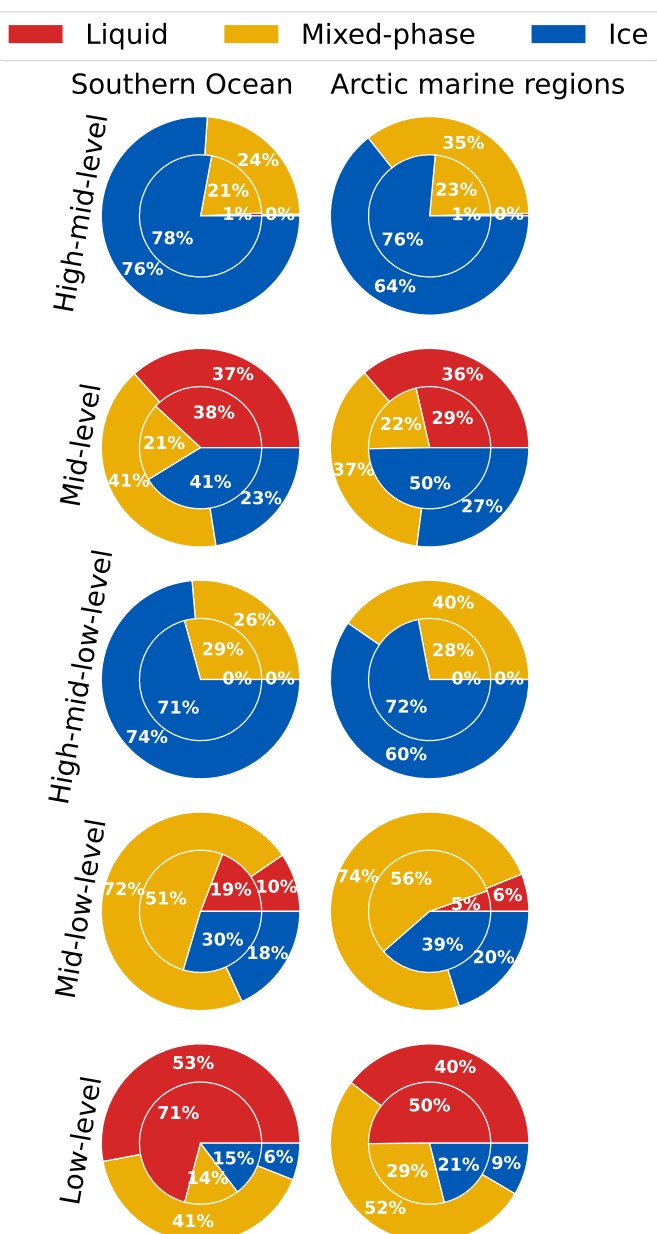

**Figure 6.** Cloud phase distributions of various cloud types (rows) over the Southern Ocean (left column) and the Arctic marine regions (right column). The outer pie charts include all cloud profiles of a cloud type, while the inner pie charts only consider cloud profiles, where the lidar signal is not fully attenuated by the cloud. The phases liquid, ice, and mixed-phase refer to the liquid fraction described in Sec. 3 and Eq. 3. Liquid corresponds to a liquid fraction of 1, ice to a liquid fraction of 0, and the mixed-phase category refers to liquid fractions larger than 0 but smaller than 1.

In general, all cloud types show quite high fractions of mixed-phase cloud profiles. The frequencies of the different phases are very similar between the Southern Ocean and the Arctic marine regions.

We don't compare the fractions of the cloud phase to previous literature (Mayer et al., 2023; Listowski et al., 2019; Huang et al., 2012) at this point, as the percentages strongly depend on the definitions and considerations of the cloud types and on the calculation and consideration of cloud phases, which vary a lot in the different studies. More detailed comparisons to previous research will be done in the following subsections investigating different aspects of the cloud phase, such as the cloud phase as a function of cloud top temperature.

### 5.2.1 Cloud phase correlation with the vertical and horizontal cloud extent

We now investigate if clouds consisting of different phases and at different levels show differences in their horizontal and vertical extent. The calculation of the horizontal and the vertical extent of the cloud objects is described in Sec. 3.

Figure 7 shows the results for the Southern Ocean and the Arctic marine regions. In general, ice clouds have a larger vertical extent, while liquid clouds show only a small vertical extent, but reach a larger horizontal extent compared to ice clouds.

Mixed-phase clouds show a broader distribution and can reach both large vertical and/or horizontal extent. In mid-low-level (ML) clouds the horizontal extent is larger for mixed-phase clouds compared to liquid clouds, both over the Southern Ocean and the Arctic marine regions. The horizontal extent of liquid clouds is smaller than in mixed-phase clouds of the same cloud type. Comparing the horizontal extent of mixed-phase and ice clouds, we can see that mixed-phase clouds have mainly a larger horizontal extent compared to ice-phase clouds, except for HML and HM clouds over the Arctic marine regions showing the opposite signal.

The results are very similar between clouds over the Southern Ocean and the Arctic marine regions. Regarding the difference between the cloud types, the vertical structure is mainly forced by the definition of the cloud types, based on cloud base heights and cloud top heights, which also constrain the maximum vertical extent. The vertical extent of mixed-phase and ice clouds is generally very similar.

The small vertical extent of liquid clouds is probably influenced by different factors. The liquid phase of the investigated clouds is presumably dominated by smaller supercooled liquid droplets. These are mainly detected by the lidar, while the radar is able to detect the liquid phase for larger droplets and rain. Therefore, the extinction of the lidar signal, due to strong attenuation, probably has a strong influence on the result of the small vertical extent of the liquid clouds. Thus, this result is at least partly coming from the limited penetration depth of the lidar signal in supercooled liquid clouds. Nevertheless, there have been studies using other observation techniques, such as ground-based remote sensing, showing that supercooled liquid cloud layers tend to be shallow (Ansmann et al., 2009).

Mixed-phase clouds show a larger vertical extent compared to liquid clouds in all cloud types and both over the Southern Ocean and the Arctic marine regions. This is also in line with theoretical knowledge about polar mixed-phase clouds. Typically, heterogeneous ice nucleation occurs in the supercooled liquid layer, the Wegener-Bergeron-Findeisen process leads to the growth of the ice crystals, which then fall from these liquid layers and form virga. Including the virga in the cloud extent, the cloud extent increases with the formation of ice compared to the supercooled liquid cloud.

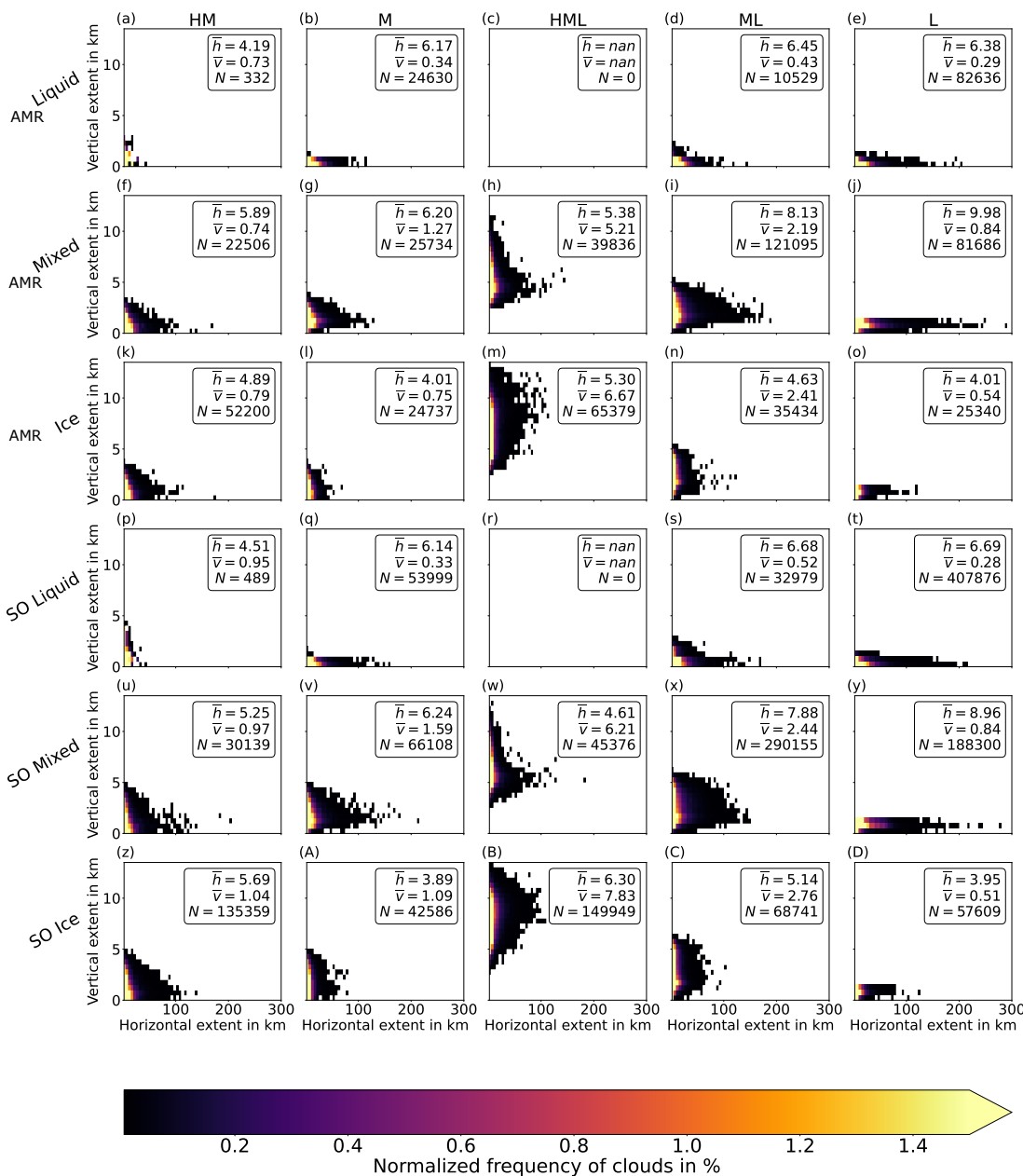

**Figure 7.** 2-dimensional histogram of clouds with different phases regarding their horizontal and vertical extent. Minimum extent are 2 vertical layers, each 60m, and 2 horizontal profiles. Cloud profiles are considered as the same cloud if the time difference to the next profile of the same cloud type is less or equal than $0.2\,\mathrm{s}$ (see. Sec.3). The unit of the horizontal extent is $\mathrm{km}$, assuming one vertical profile having a horizontal distance of about $1.1\,\mathrm{km}$ to the next profile. $\overline{h}$ describes the mean horizontal extent in number of profiles, $\overline{v}$ describes the mean vertical extent. Top 3 rows: Arctic marine regions (AMR), bottom 3 rows: Southern Ocean (SO).

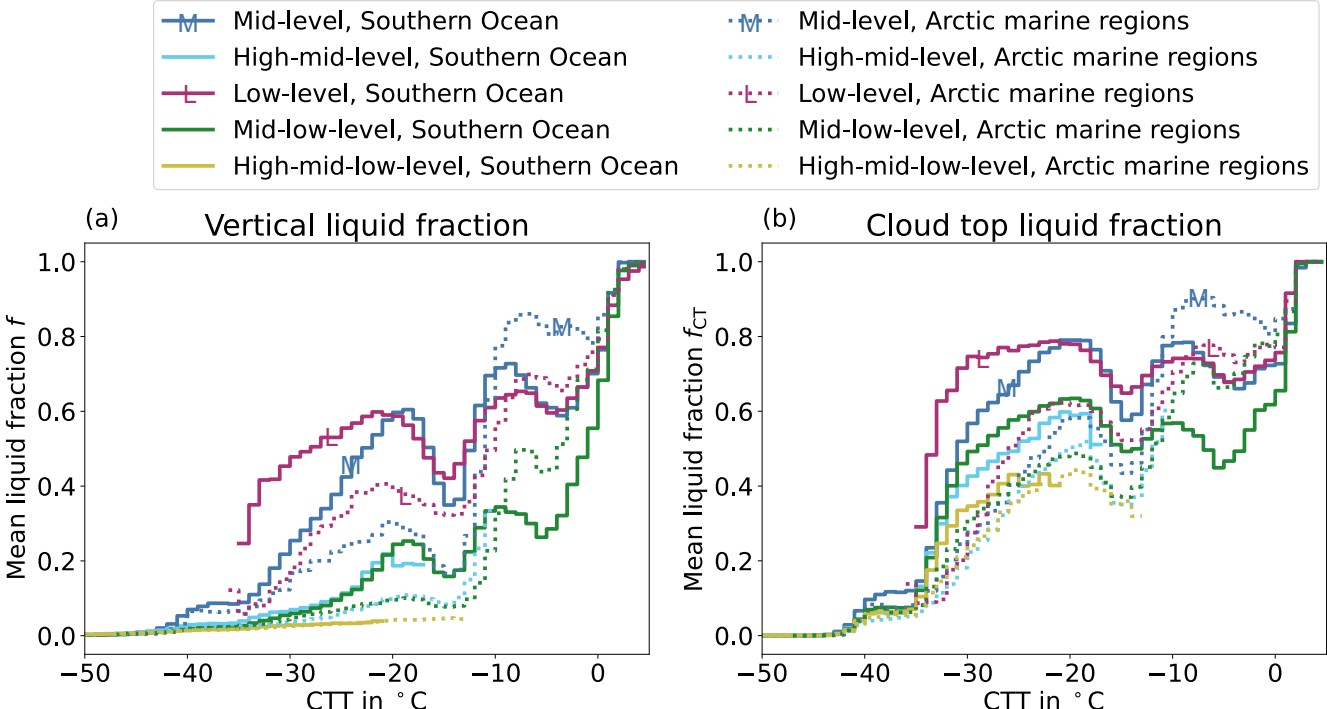

**Figure 8.** Mean liquid fraction of profiles of different cloud types as a function of the cloud top temperatures (CTT). In panel (a) the liquid fraction $f$ is calculated for each vertical cloud column and then averaged for each $1\,°C$ bin of the CTT. The liquid fraction of each profile is calculated as described in Sec. 3, Eq. 3. In panel (b) only the cloud top phase is considered and the liquid fraction $f_{CT}$ is calculated as described in Eq. 4, but only using the phase of all cloud top bins of one specific cloud type.

Zhang et al. (2014) investigated the spatial scales of altocumulus clouds with globally collocated CloudSat/CALIPSO observations and found a vertical extent of $1.96\,km$ ($\pm\,1.10\,km$) and a horizontal extent of $40.2\,km$ ($\pm\,52.3\,km$). The results for the vertical extension matches quite well, with our result for mid-level clouds being slightly smaller, while the mean horizontal extent in our analysis is smaller ($3.89\,km$-$6.24\,km$). However, the horizontal extent is strongly influenced by excluding specific cases like multi-layer cloud profiles, which could lead to shorter horizontal scales, due to an overlap with a different cloud

layer. Other uncertainties occur in clouds having CTH and CBH close to the thresholds of the cloud type definition, which for example can lead to a cloud being partly classified as low-level and partly as mid-low-level by increasing CTH. This cloud would be considered as two separate clouds, as they aren't considered as the same cloud type.

### 5.2.2 Cloud phase dependence on the cloud top temperature

We now investigate how the cloud phase correlates with the cloud top temperature, especially in the mixed-phase temperature regime between -38 °C and 0 °C. Figure 8 shows the cloud phase as a function of the CTT for the different cloud types in panel (a), and the liquid fraction of cloud top phase in panel (b), as it was explained in Sec. 3.

First of all, we can see the high liquid fraction in low-level clouds especially over the Southern Ocean at relatively low temperatures between -40 °C and -17 °C in Fig. 8. Mid-level clouds show high liquid fractions as well, while mid-low-level clouds show rather small liquid fractions. A small liquid fraction can also be observed for high-mid-level clouds, with high-mid-level clouds occurring at lower temperature compared to mid-low-level clouds. Regarding the cloud top liquid fraction $f_{\mathrm{CT}}$ in panel (b) in Fig. 8, the liquid fraction increases for all cloud types compared to the vertical liquid fraction $f$, indicating a preferential occurrence of liquid at the cloud top.

Thinner cloud layers (L,M) have a higher liquid fraction compared to thicker cloud layers (HM, ML, HML) extending over several troposphere layers, which was already described in Sec. 5.2.1. We can even see this, if we only consider the cloud top phase, where the uncertainty introduced by the lidar extinction has no influence.

In general, the liquid fractions of clouds over the Southern Ocean are higher compared to the liquid fractions of clouds over the Arctic marine regions for CTT $< -10$ °C, while for high temperatures (CTT $> -10$ °C) the liquid fraction in clouds over the Arctic marine regions is larger compared to the Southern Ocean.

Further important features are the local minima in the liquid fraction mainly seen at a CTT around -15 °C and partly around -5 °C. Interestingly, this feature has already been seen in many other studies (Nagao and Suzuki, 2022; Danker et al., 2022; Zhang et al., 2019; Alexander and Protat, 2018; Zhang et al., 2014; Riley and Mapes, 2009), but has only partly been investigated and sometimes not even described. Zhang et al. (2014) saw a similar peak in the mixed-phase fraction at -15 °C, investigating the vertical and horizontal scales of Ac clouds, but focused more on the differences between different regions. Danker et al. (2022) also showed the phenomena of increased ice formation at -15 °C and -5 °C and discussed many other studies, which have shown this behaviour or which explain possible reasons. In the following, we will mention a few of them.

Riley and Mapes (2009) found an unexpected peak at -15 °C in the CloudSat echo, but studies have also shown the peak using lidar observations (Nagao and Suzuki, 2022), which hints that this effect is not due to an issue of a specific instrument. Regarding this temperature range, there are various studies describing different processes occurring around these temperatures, maybe even interacting and causing other processes. The ice habits strongly depend on the temperature, with column-needles occurring/growing at temperatures around -5 °C and plate-stellar dendrites around temperatures of -15 °C (Avramov and Harrington, 2010; Fukuta and Takahashi, 1999). Especially, the dendritic growth zone occurring at -15 °C has also been investigated by other studies (von Terzi et al., 2022; Silber et al., 2021) and might lead to this increased ice fraction. The strong growth of the ice crystals around these temperatures also leads to an increased signal in the remote sensing instruments and therefore to an increased detection of ice. Another process correlating with dendritic ice crystals is an increased aggregation rate (Chellini et al., 2022), but also the possibility of secondary ice formation (Mignani et al., 2019; Sullivan et al., 2018) due to small fragments in case of collisions.

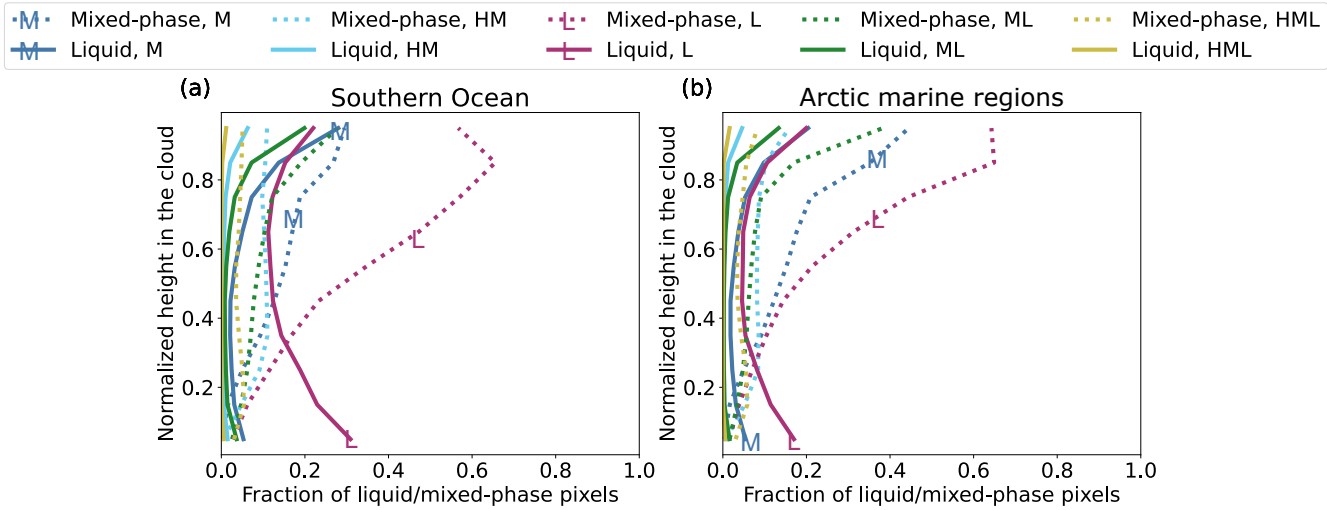

**Figure 9.** Phase fraction at normalised cloud heights for mixed-phase clouds. Cloud types over the Southern Ocean are shown in panel a), cloud types over the Arctic marine regions in panel b). The solid lines show the fraction of liquid vertical bins at certain normalised heights, while the dotted lines show the fraction of mixed-phase vertical bins. The normalised height at the y axis is from 0 to 1 and values are calculated in steps of 0.1.

All of these processes might play a role as they affect each other. From the satellite perspective it is hard to pinpoint the increased ice fraction to specific processes, because it could even be a combination of processes and the relevance of processes might also vary depending on a specific region, time, and the conditions there. Nevertheless, the potential to even see these
increased ice productions at specific temperatures is not yet fully exploited. The combination of this knowledge with other ground based observations or laboratory experiments could improve our understanding of the cloud phase. This improved process understanding can also lead to a better representation in models, which contrary to our results show a rather smooth phase partitioning (e.g. McCoy et al., 2016).

### 5.2.3   Vertical phase distribution of mixed-phase clouds

We now investigate how the phase is distributed vertically within the clouds. For this purpose, we limit our analysis to mixed-phase clouds, as pure ice and pure liquid cloud profiles don't show a vertical phase distribution. We only consider cloud profiles, where the lidar is not fully attenuated to reduce the resulting uncertainties in the vertical phase distribution. To investigate the vertical phase distribution of mixed-phase clouds we analyse all cloud profiles in which either both liquid vertical bins and ice vertical bins are observed, or any mixed-phase vertical bins are observed in one vertical cloud column. Normalising the height
within the cloud, we calculate the fraction of liquid vertical bins at specific heights within the clouds, as well as the fraction of mixed-phase vertical bins (see Fig. 9).

Most of the mixed-phase clouds show an increased liquid fraction at cloud top, and it strongly decreases towards lower heights within the cloud, except for low-level clouds which show similar liquid fractions at lower heights of the cloud. Further investigations (not shown) of low-level clouds have shown that this might mainly be based on the category 'Multiple scattering due to supercooled liquid' (see Tab. 1). Most cloud types show also an increase of the liquid fraction at cloud base. Both, the increased liquid fraction at cloud base, and the generally high liquid fraction in low-level clouds at lower heights persist, if we restrict our analysis to cloud profiles with a maximum temperature of $0\,°C$, and are therefore not related to melting. So far, it is not yet clear if the increased liquid fraction at lower heights is based on uncertainties like ground clutter or actually due to more liquid water in clouds and further research is needed.

The maximum of the fraction of mixed-phase vertical bins is located slightly below cloud top and most of the mixed-phase vertical bins are located in the upper half of the cloud. The structure of mixed-phase clouds with an increased liquid fraction at cloud top has been already seen in many other observations (Zhang et al., 2019; Carey et al., 2008; Fleishauer et al., 2002). Studies based on ground-based remote sensing instruments also observed liquid cloud tops with ice mainly below in mixed-phase clouds (Zhang et al., 2017; Kalesse et al., 2016; de Boer et al., 2011).

The increased fraction of liquid and mixed-phase bins at cloud top is in line with the results found in Sec. 5.2.2 showing an increased cloud top liquid fraction compared to the vertical liquid fraction. The high frequency of supercooled liquid at cloud top was also shown by Schima et al. (2022) using airborne radar, lidar, and in-situ measurements collected during the Southern Ocean Clouds, Radiation, Aerosol Transport Experimental Study (SOCRATES).

### 5.2.4 Cloud phase as a function of the sea ice concentration and the aerosol concentration

To investigate the differences in cloud phase as a function of the sea ice concentration, the mean liquid fraction of clouds over the open ocean are compared with the mean liquid fraction of clouds over sea ice. The detailed method is described in Sec. 3. Figure 10 shows the significant differences between the mean liquid fractions in clouds over ocean and the mean liquid fraction in clouds over sea ice for different cloud types as a function of the cloud top temperature. It's important to address the behaviour at different temperatures, otherwise the generally lower temperatures over sea ice will naturally lead to a higher proportion of ice clouds.

Over the Southern Ocean the liquid fraction of clouds over sea ice is significantly higher compared to the clouds over the open ocean especially in low-level clouds, but also in mid-level and mid-low-level clouds. Over the Arctic marine regions we can see the same behaviour for low-level clouds, but mid-level and mid-low-level clouds only show small signals for cloud top temperatures warmer than -10 °C. Furthermore, the difference of the liquid fractions over the Southern Ocean is higher in low-level and mid-level clouds, compared to mid-low-level clouds. A similar result for low-level clouds has been found by Carlsen and David (2022) using a similar dataset based on CloudSat and CALIPSO observations, but a different metric for the cloud phase. The hypothesis proposed in Carlsen and David (2022) is that due to the coverage of the ocean by sea ice, less sea spray aerosols can be released, which leads to less INPs over sea ice and thereby a higher fraction of liquid clouds.

Comparing CAMS reanalysis data in clouds over sea ice and in clouds over the ocean Fig. 11 shows a systematic difference in the sea salt concentration over sea ice and over open ocean with less sea salt over sea ice, which supports the hypothesis of

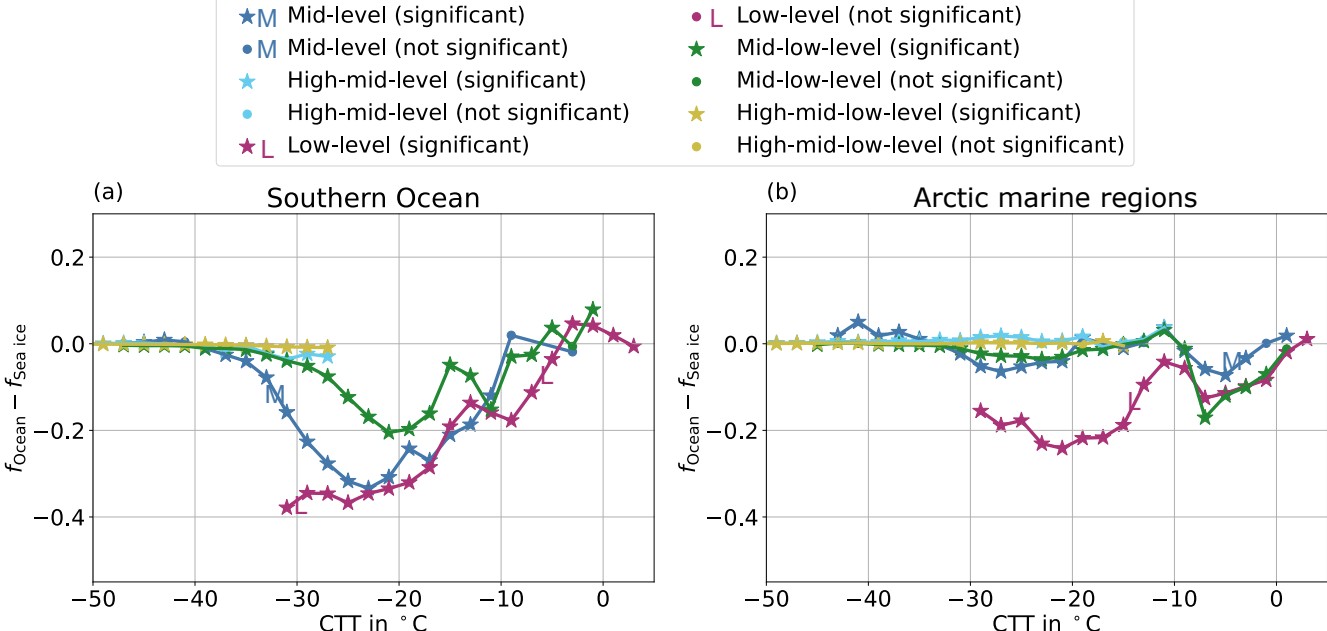

**Figure 10.** Difference between the mean liquid fraction of clouds over ocean and the mean liquid fraction of clouds over sea ice as a function of the cloud top temperature. The significance of the two distributions of liquid fractions of clouds over ocean and clouds over sea ice is investigated using a Z-test with a p-value of 0.05 for each CTT-bin. Panel a) shows the results for the Southern Ocean, panel b) shows results for the Arctic marine regions. Data are only shown if there are at least 500 cloud profiles over ocean and 500 profiles over sea ice.

Carlsen and David (2022). Sea salt acts hereby as a proxy for sea spray aerosols, as the ice nucleating part of the aerosols are usually biological components from the sea surface microlayer, like microorganisms acting as INP at high temperatures (Porter et al., 2022; Burrows et al., 2013; Després et al., 2012). Figure 11 also shows that in mid-level clouds over the Arctic sea ice, the mixing ratio of sea salt is slightly lower than the mixing ratio of organic matter, while in mid-level clouds over the Southern Ocean, sea salt shows much higher values compared to other aerosol types. The role of other aerosol types than sea salt may be higher in the Arctic compared to the Southern Ocean. This would also explain the missing correlation of sea ice with these clouds, which is shown in Fig. 10 (right panel, blue line). Furthermore, the transport of other aerosol types like dust is more important in the Arctic compared to the Southern Ocean. As the long-range transport occurs usually above the boundary layer, this is more relevant for mid-level or mid-low-level clouds or even higher clouds and might also reduce the effect of locally emitted sea spray aerosols for these clouds.

Nevertheless, a recent study of Papakonstantinou-Presvelou et al. (2022) investigated ice number concentrations in Arctic boundary layer ice clouds and found a higher ice number concentration in clouds over sea ice compared to clouds over ocean, especially in the latitudes between $60\,°$ N and $70\,°$ N and for temperatures between -10 °C and 0 °C, which may seem contradictory to our results of a lower liquid fraction over ocean and the assumption of less INPs over sea ice. The results

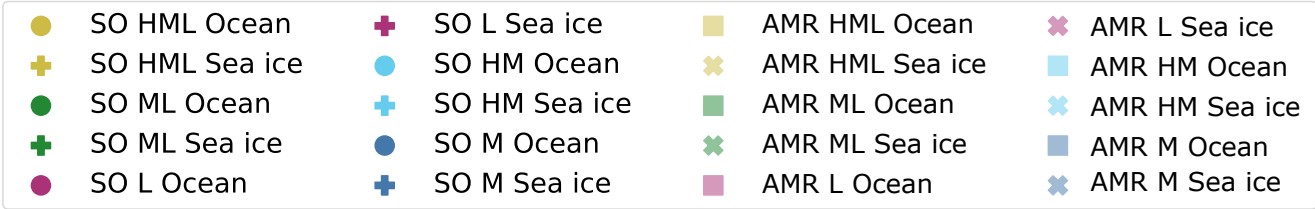

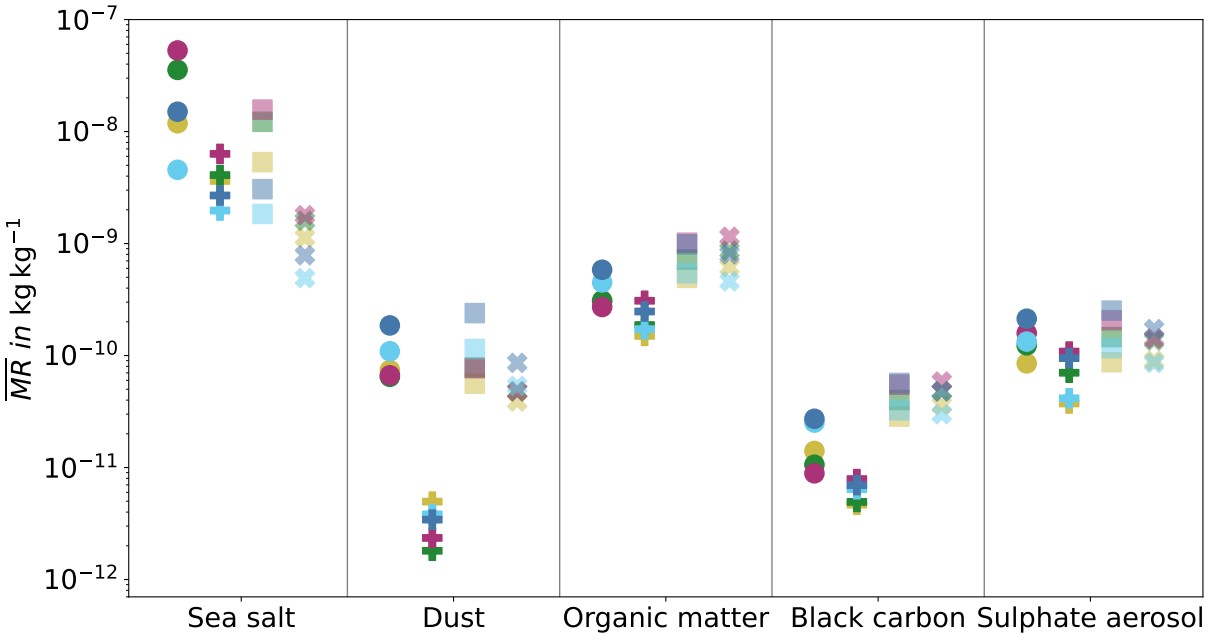

**Figure 11.** Vertically averaged aerosol mixing ratios (MR) within the heights of a cloud column for different aerosol types. Shown is the mean for different categories, namely the Southern Ocean (SO - darker colours), Arctic marine regions (AMR - lighter colours), clouds over sea ice (dots and squares) and clouds over the open ocean (plus signs and crosses), and various cloud types (colours).

of Papakonstantinou-Presvelou et al. (2022) may be influenced by secondary ice production caused by blowing snow particles, which is dependent on the wind velocity and is especially relevant for low-level clouds close to the ground. In addition, Papakonstantinou-Presvelou et al. (2022) investigate only ice clouds, while our study investigates the general cloud phase. Furthermore, Papakonstantinou-Presvelou et al. (2022) found the strongest difference in ice number concentrations at cloud top temperatures larger than -10 °C, while we found the strongest difference of cloud phase over open ocean and cloud phase

over sea ice at lower temperatures. Nevertheless, the discrepancy of the different results show the need of further research to improve our understanding of the processes most relevant for cloud phase in remote regions.

     We now investigate the correlation of the aerosol concentrations with the cloud phase. Figure 12 shows that low-level, mid-low-level and mid-level clouds show a stronger correlation with aerosol concentrations compared to high-mid-level clouds, which show a small signal over the Southern Ocean. High-mid-low-level clouds show no signal, possibly because they mostly

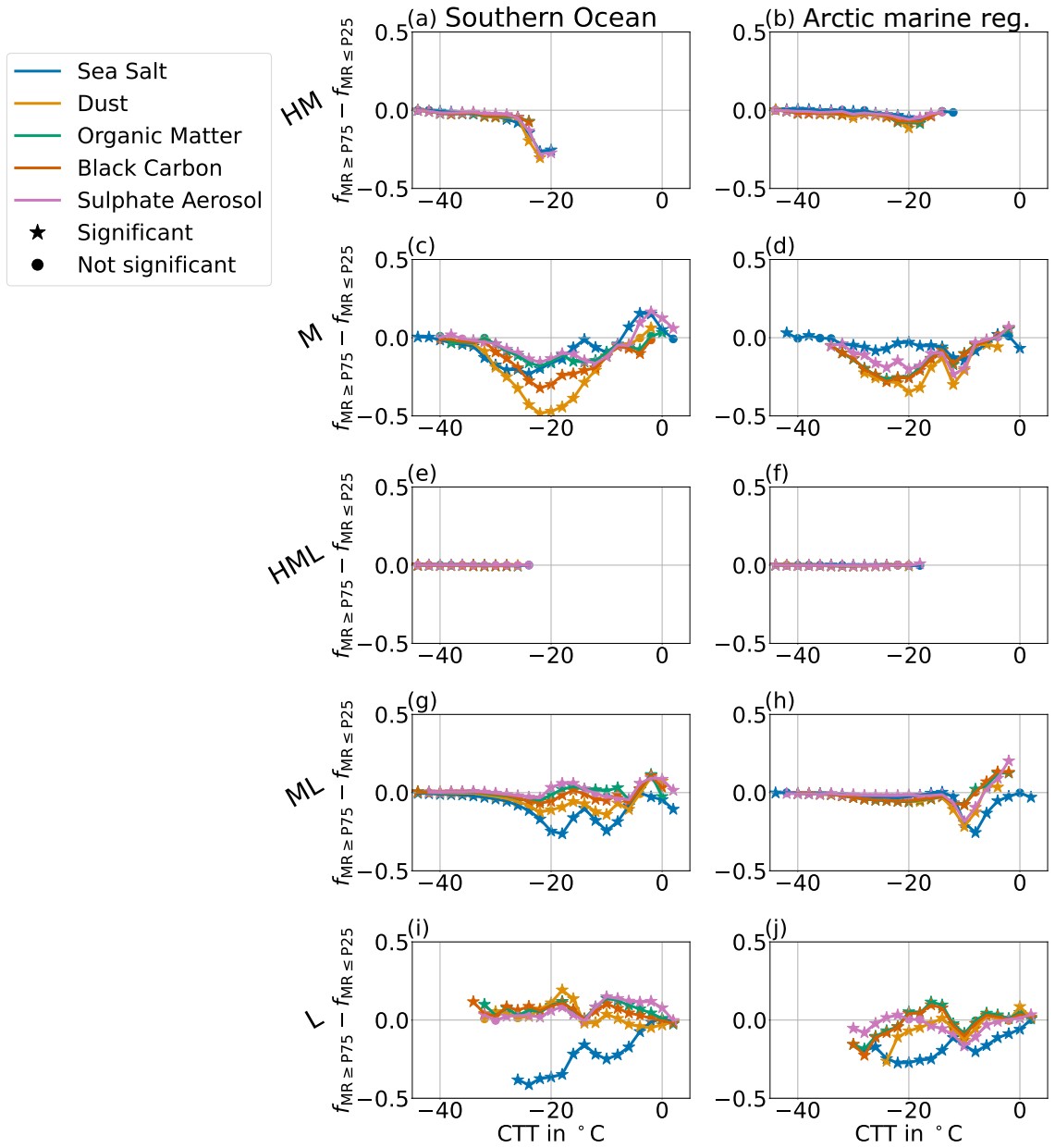

**Figure 12.** Difference of the mean liquid fraction of clouds collocated with high mixing ratio of an aerosol type, larger than the 75th percentile, and the mean liquid fraction of a cloud collocated with low aerosol mixing ratio, lower than the 25th percentile. Negative values correspond to an increased ice fraction with higher aerosol concentrations. A Z-test with a p-value of 0.05 is used to investigate the significance of the difference. Data are only shown if there are at least 500 cloud profiles with low aerosol concentrations and 500 cloud profiles with high aerosol concentrations.

consist of ice (see Fig. 8), but the attenuation of the remote sensing signals may introduce uncertainties here. The negative values in general show the decrease of the liquid fraction with high aerosol concentration and thereby a higher ice fraction, which is in line with the assumption of additional aerosols acting as additional INPs. Furthermore, the liquid fraction in low-level clouds over the Southern Ocean is lower in high sea salt conditions compared to low sea salt conditions. The content of other aerosol types only shows small changes in the liquid fraction (compare panel (i) in Fig. 12). Note, that we interpret sea

salt here as a proxy for sea spray aerosols including biological parts, which can act as INP. The liquid fraction in low-level clouds over the Arctic marine regions is lower in high sea salt conditions, but there, other aerosol types may play a role (panel (j)). This could be explained by the fact that other aerosol sources are much closer and the transport plays probably a larger role there, compared to the very remote Southern Ocean. In mid-low-level clouds, sea salt aerosols show the largest differences in the liquid fraction between low and high aerosol conditions, but a generally smaller difference in the liquid fraction based

on the sea salt conditions compared to low-level clouds.

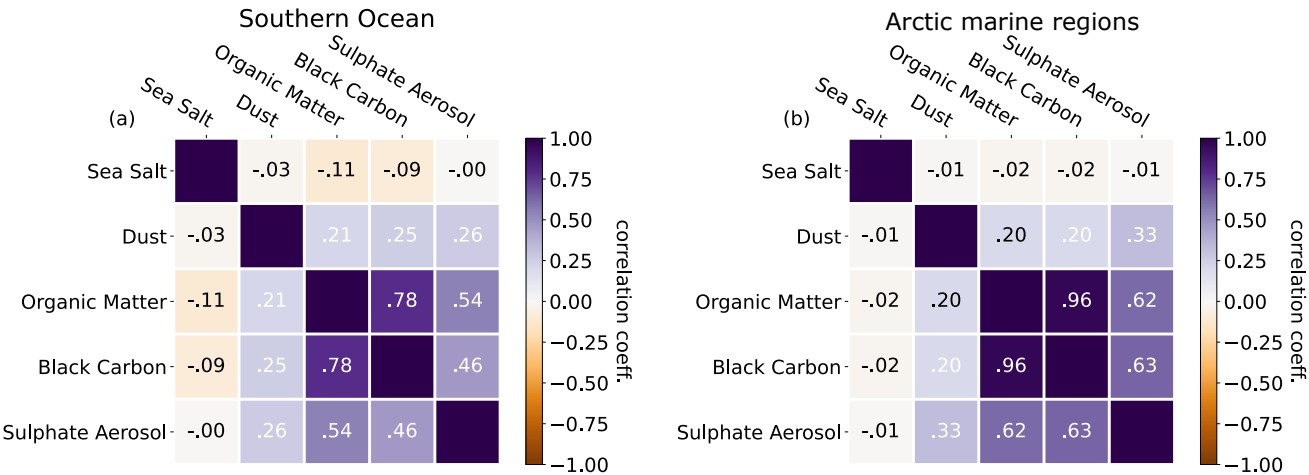

**Figure 13.** Pearson's correlation coefficient of the mean mixing ratios of different aerosol categories within a cloud. Left panel shows results for the Southern Ocean, right panel for the Arctic marine regions.

In mid-level clouds over the Southern Ocean dust concentrations seem to play a large role, but also other aerosol types like black carbon or sea salt correlate with the liquid fraction. Over the Arctic marine regions mid-level clouds are as well influenced by many aerosol types and sea salt may play a minor role there. Interestingly, the correlation of sea salt, as a proxy for sea spray, with the cloud phase matches quite well with the correlation of sea ice with the phase regarding the different

cloud types (compare Fig. 10). This supports the hypothesis of Carlsen and David (2022) that sea spray particles acting as INP foster the glaciation of low-level clouds.

Additional support for the hypothesis that sea spray aerosol strongly impact the phase of polar low clouds is given by the work of Griesche et al. (2021), who found a dependence of the Arctic cloud phase on the surface coupling, connected with marine INPs, but only for temperatures warmer than -15 °C, while we see a correlation of sea salt aerosol with the cloud

phase for much lower temperatures. Griesche et al. (2021) observed a higher frequency of ice containing clouds, if the clouds are coupled to the surface compared to decoupled clouds using radiosonde data, and ground-based lidar and radar data. They propose increased marine biological INPs in the surface coupled boundary layer as the reason on the basis of recent in situ INP measurements in the Arctic. As this study takes place during the Arctic summer, temperatures close to the surface are rather high, compared to the winter season. Since our study uses a two-year dataset, we assume that the influence of these marine biological INPs is also relevant at lower temperatures compared to Griesche et al. (2021). The relevant factor is probably the proximity of the cloud to the surface. This would also explain, why we see an influence of sea salt on the cloud phase at lower temperatures compared to Griesche et al. (2021).

To make sure that the phase influence is based on the aerosol concentration of a specific aerosol type and not just based on correlations between different aerosol types, we calculated the correlations coefficients of the different aerosol types within the different cloud types (see Fig. 13). Regarding correlations with sea salt, we can clearly see that there are only very low correlation coefficients with a most negative value of -0.11. Highest correlations (0.96) can be seen between organic matter and black carbon in the Arctic marine regions, but also over the Southern Ocean (0.78). Sulphate also correlates partly with organic matter or black carbon (0.46 - 0.63). Dust shows generally lower correlations with other aerosol types, with maximum correlations of 0.33 with sulphate aerosol. This strengthens the previously described hypothesis that sea spray seems to be an important INP in low-level and mid-low-level clouds with high concentrations leading to a reduced liquid fraction, and similar for dust in mid-level clouds.

## 5.3  Cloud radiative effect

We now investigate the cloud radiative effect (CRE) of the different cloud types and examine the influence of the cloud phase on the CRE. We further investigate the contribution of various cloud types to the total CRE over the Southern Ocean and the Arctic marine regions. To make the incoming solar radiation comparable between the Southern Ocean and the Arctic marine regions, we use equal latitude bands in this section for the two regions, namely $60\,°S/N$ to $82\,°S/N$. In all other sections, the Southern Ocean is defined from $40\,°S$ to $82\,°S$, as it is described in the Sec. 3. Nevertheless, there are still biases due to the different land distribution over the Southern Ocean and the Arctic marine regions, which leads to a higher cloud frequency in high latitudes over the Arctic marine regions compared to the Southern Ocean, because land surfaces like the Antarctic continent are excluded. For this analysis we consider both cloud profiles over the open ocean and cloud profiles over sea ice.

### 5.3.1  Mean cloud radiative effect of different cloud types

Panels (a) and (b) in Fig. 14 show the mean shortwave CRE (SWCRE), the longwave CRE (LWCRE), and the net CRE (NETCRE) of the different cloud types. High-mid-low-level and mid-low-level clouds show the highest SWCRE over the Southern Ocean, while over the Arctic marine regions mid-level clouds have a similar SWCREs compared to high-mid-low-level and mid-low-level clouds over the Arctic marine regions. Over the Southern Ocean the SWCRE is generally stronger compared to the results from Oreopoulos et al. (2017), who investigated the SWCRE of similar cloud types globally with the same dataset, but Oreopoulos et al. (2017) also found the strongest SWCRE for high-mid-low-level clouds. Over the Arctic

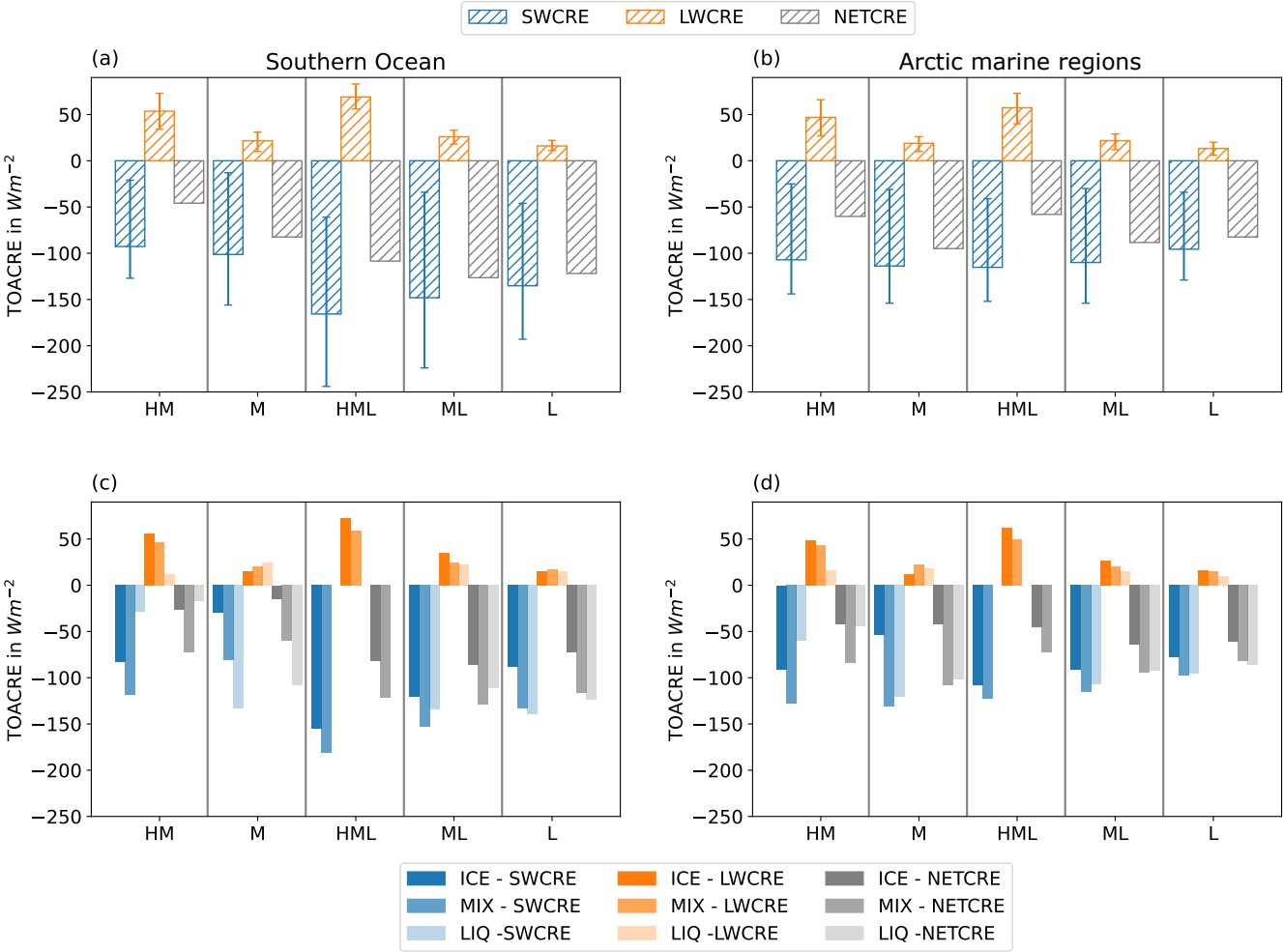

**Figure 14.** Cloud radiative effects of different cloud types and with different cloud phases. The upper row shows the mean of the top of the atmosphere cloud radiative effect (TOACRE) for different cloud types, with the error bars showing the 25th and 75th percentiles. The lower row shows the mean top of the atmosphere cloud radiative effects (TOACRE) for various cloud types as a function of cloud phase. Shortwave cloud radiative effect (SWCRE) is shown in blue colours, longwave cloud radiative effect (LWCRE) is shown in orange colours, and net cloud radiative effect (NETCRE) in grey colours. The left column (panel (a) and (c)) shows the results for the Southern Ocean, while the right column shows the results for the Arctic marine regions (panel (b) and (d)). Note that there are no bars for liquid high-mid-low-level clouds in panel (c) and (d), as they don't occur.

marine regions, the SWCRE of the different cloud types is very similar and doesn't vary much. We find a generally stronger SWCRE over the Arctic marine regions compared to the global average from Oreopoulos et al. (2017), except for high-mid-low-level clouds over the Arctic marine regions. We can see that high-mid-low-level, mid-low-level and low-level clouds show a higher SWCRE over the Southern Ocean compared to the Arctic marine regions. A possible reason could be the distribution

of sea ice, as the SWCRE is lower for clouds over sea ice compared to clouds over open ocean. The fraction of observed cloud profiles over sea ice is larger over the Arctic marine regions compared to the Southern Ocean due to the different land distribution in the two hemispheres. Contrarily, high-mid-level and mid-level clouds have a higher SWCRE over the Arctic marine regions compared to the Southern Ocean. Further analysis shows that this difference persists when we only consider cloud profiles over ocean and analyse the SWCRE as a function of the latitude (not shown). This hints at cloud property differences over the Arctic marine regions and the Southern Ocean such as vertical extent but also optical properties which may be related to aerosols. In both regions high-mid-low-level clouds and high-mid-level clouds show the highest LWCRE. The LWCRE are similar over the Arctic marine regions and the Southern Ocean, with slightly larger effects over the Southern Ocean. McFarquhar et al. (2021) showed a lower SWCRE and a higher LWCRE of low clouds over the Southern Ocean observed during SOCRATES compared to our results, but differences in cloud type definitions, seasons, and regions might introduce large uncertainties to the comparison of averaged values.

### 5.3.2 Dependence of the cloud radiative effect on the cloud phase

The phase of clouds has a large effect on their CRE, because numerous small liquid droplets are optically thicker than (few large) ice particles. Therefore, we investigate now the influence of the cloud phase on the CRE, shown in panel (c) and (d) in Fig. 14.

The highest SWCRE in all cloud types is observed from mixed-phase clouds, except for mid-level and low-level clouds over the Southern Ocean (see second column in panel (c), Fig. 14), where liquid clouds show a higher SWCRE compared to mixed-phase clouds. Matus and L'Ecuyer (2017) investigated the global TOACRE of different cloud phases with the same dataset, but didn't distinguish different cloud types and found the general highest SWCRE for liquid clouds, which highlights the complexity and dependence of the SWCRE on the region, the cloud types and their optical properties. Comparing the SWCREs of ice and liquid clouds, we can see that in most cloud types, the SWCRE is higher for liquid clouds compared to ice clouds, except for high-mid-level clouds, where ice clouds show a higher SWCRE. The behaviour of the high-mid-level clouds was unexpected, because cloud layers consisting of liquid droplets are in general optically thicker and should therefore show a stronger negative SWCRE compared to cloud layers containing ice particles, which are optically thinner. The stronger SWCRE of high-mid-level ice clouds compared to liquid clouds is probably explained by the larger vertical extent of ice clouds. Furthermore, the different vertical resolutions of the CloudSat radar and the CALIOP lidar can have an influence on the calculation of the vertical thickness and the calculated CREs. Oreopoulos et al. (2017) shows the strong correlation of the vertical thickness on the TOA SWCRE. This is probably also the reason, why mixed-phase clouds show mainly a stronger SWCRE compared to liquid clouds. The strong extinction of the lidar signal leads probably to an underestimation of the vertical thickness of the liquid clouds. Nevertheless, liquid clouds usually are not as vertically thick as ice clouds. Therefore, the possible underestimation of the vertical extent due to lidar extinction is not the main reason for the lower SWCRE in liquid clouds compared to mixed-phase clouds. It is rather the larger vertical extent in mixed-phase clouds due to precipitating ice virga and a thereby increased optical thickness.

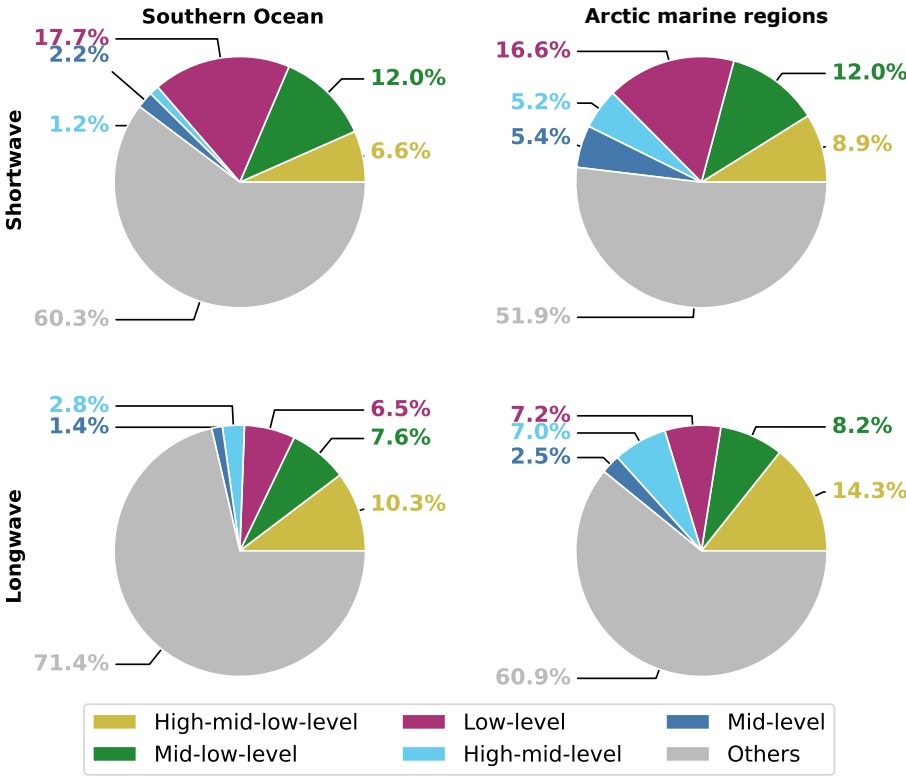

**Figure 15.** Contribution of different cloud types to the total cloud radiative effects over 2 years at the top of the atmosphere over the Southern Ocean and the Arctic marine regions in percent.

Regarding the differences between the Southern Ocean and the Arctic marine regions, it can be seen that both the SWCRE and the LWCRE are mainly larger over the Southern Ocean compared to the Arctic marine regions. It can be also seen that over the Southern Ocean the SWCRE of clouds with low cloud bases (HML, ML, L) are quite high, while over the Arctic marine regions clouds with middle cloud base heights (HM, M) show similar SWCRE compared to clouds with low clouds bases. Generally, it can be seen that the cloud phase, but also the vertical extent of the cloud, has a large influence on the CREs of the different cloud types.

### 5.3.3 Contribution of different cloud types to the total SWCRE and LWCRE

To investigate the contribution of the different cloud types to the total SWCRE and LWCRE, we calculate the total sum of the SWCRE and the total sum of the LWCRE over the full two years 2007 and 2008 and normalise it by the total number of observed cloud profiles. The numbers are shown in Tab. A1. We also calculate the total sum of the SWCRE and LWCRE of different cloud types. Figure 15 shows the contribution of the different cloud types to the total SWCRE and LWCRE as a

percentage and indicates the large contribution of the low-level clouds with 17.7 % in the Southern Ocean and 16.6 % in the Arctic marine regions to the total SWCRE. This is related to their high occurrence shown in Fig. 4 and similarly shown by McFarquhar et al. (2021). Besides low-level clouds, mid-low-level and high-mid-low-level clouds show a large contribution to the SWCRE, but show higher LWCRE, compensating SWCRE. Mid-level clouds only contribute to a minor role to the total SWCRE with 2.2 % in the Southern Ocean, but show a higher contribution of 5.4 % in the Arctic marine regions. The contribution to the LWCRE of mid-level clouds over the Southern Ocean is 1.4 % and slightly larger over the Arctic marine regions with 2.5 %. The contribution of high-mid-level clouds to the SWCRE is small (1.2 %) over the Southern Ocean, but larger over the Arctic marine regions (5.2 %) with LWCRE contribution of 2.8 % over the Southern Ocean and 7.0 % over the Arctic marine regions.

Regarding the total NETCRE (not shown), low-level clouds have the largest (negative) effect followed by mid-low-level clouds.

In summary, from an examination of various aspects of the cloud radiative effect, we see that mid-low-level clouds over the Southern Ocean have the highest net CRE, while over the Arctic marine regions mid-level clouds have the highest net CRE (see Fig. 14). In general mixed-phase clouds show a more negative SWCRE compared to ice and liquid clouds except for mid-level and low-level clouds over the Southern Ocean. Therefore, mixed-phase and liquid clouds show a higher net CRE compared to ice clouds. In general clouds over the Southern Ocean show a higher CRE compared to clouds over the Arctic marine regions. Investigating the contribution of different cloud types to the total CRE over the Southern Ocean and the Arctic marine regions, the low-level clouds contribute most in both regions due to their higher frequency compared to other cloud types (see Fig. 15).

## 6 Conclusions

Climate models struggle to correctly simulate the cloud phases, which leads to radiative biases, especially over the Southern Ocean and the Arctic. We investigated two years of the cloud phase over the Southern Ocean and the Arctic marine regions with the DARDAR dataset based on CloudSat and CALIOP observations. We focus on clouds having low or middle cloud base heights, as these clouds are often misrepresented in climate models. High-mid-low-level, high-mid-level, mid-low-level, mid-level, and low-level cloud types are defined by their cloud base height and their cloud top height. We further used various datasets for the cloud radiative effect (CRE) (2B-FLXHR-LIDAR), sea ice concentration (Cavalieri et al., 1996), and aerosol reanalysis (CAMS) to investigate their possible influence and connections to cloud phase. Our findings are summarised below:

– **Cloud type occurrence:**

High frequencies of low-level clouds can be observed over the Southern Ocean (15.8 %) while over the Arctic marine regions low-level clouds occur in 8.6 % of all observed profiles. Mid-low-level clouds occur in more than 5 % of the profiles. High-mid-low-level clouds occur in about 3.3 - 3.5 % of the profiles. Mid-level or high-mid-level clouds are found in 2.2-2.6 % of the observations, but regarding uncertainties due to the wrong representation of the cloud phase, mid-level clouds together with mid-low-level and low-level clouds are most important, as they occur in the mixed-phase temperature regime in these regions.

- **Cloud phase:**

  - All investigated cloud types show a high fraction ($\geq 24\,\%$) of mixed-phase clouds.

  - Cloud phase strongly correlates with the vertical and the horizontal extent of the cloud. Liquid clouds are usually vertically thinner, but have a larger horizontal extent compared to ice clouds except for liquid high-mid-level clouds, which are horizontally larger compared to ice-phase high-mid-level clouds. Mixed-phase clouds show generally a high vertical and horizontal extent.

  - The liquid fraction strongly decreases between -10 °C and -15 °C and around -5 °C, which hints to ice formation processes occurring in these clouds, especially at these temperature ranges. Habit dependent vapour growth or secondary ice production are discussed as possible processes being the reason, but further laboratory experiments and simulations are needed to verify.

  - Higher liquid fractions in clouds over sea ice compared to the open ocean have been found in low-level, mid-level, and mid-low-level clouds over the Southern Ocean. The same behaviour has been seen in low-level clouds over the
  Arctic marine regions. The hypothesis of Carlsen and David (2022) explains this with the prevention of the release of sea spray aerosol, acting as INP, due to sea ice coverage. The investigation of the aerosol content with CAMS reanalysis, especially sea salt as a proxy for sea spray aerosols acting as INP, supports the hypothesis. We could see lower sea salt concentration in clouds over sea ice compared to clouds over the ocean, particularly in low-level
  clouds. In higher altitudes, the relevance of other aerosol types like organic matter or dust increases, probably related to aerosol transport. In mid-level clouds differences in dust conditions coincide with the largest difference in the liquid fraction compared to other aerosol types.

  - We found in general higher liquid fractions in clouds over the Southern Ocean compared to the Arctic marine regions for cloud top temperatures colder than -10 °C, while for warmer cloud top temperatures than -10 °C, clouds
  in the Arctic marine regions show a higher liquid fraction than clouds over the Southern Ocean. The reason for this is still an open question and should be investigated further in future studies. Possible influences might be specific aerosols occurring in the different regions and acting as INP at different temperatures, but also other influences like dynamics have to be investigated in future research.

  - The liquid phase mainly occurs at cloud top in mixed-phase clouds with mixed-phase vertical bins in the upper half
  of the cloud and precipitating ice below, which is the typical mixed-phase cloud structure in polar regions. Only low-level clouds show also liquid phase in lower parts of the cloud due to multiple scattering.

- **Cloud radiative effect (CRE):**
  Regarding the CRE of the different cloud types, high-mid-low-level, and mid-low-level clouds show the highest short-wave CRE over the Southern Ocean, while the CREs of the different cloud types over the Arctic marine regions are in
  a more similar range. Nevertheless, including the occurrence of the different cloud types, low-level clouds clearly show the largest contribution to the shortwave CRE in both polar regions followed by mid-low-level clouds.

Overall, cloud properties are remarkably similar between the two investigated regions on the Northern and Southern hemispheres, implying that they are governed by the same cloud physical processes despite differences in meteorological and surface conditions. In future studies the results of this systematic analysis should be compared with model simulations to identify differences and improve the representation of these clouds in climate and weather models. An improved representation of the mid-low-level, low-level, and mid-level cloud phase over the Southern Ocean and Arctic marine regions is needed in weather and climate models to reduce the uncertainties in the radiative balance. In order to provide a more reasonable representation of cloud phase, a more realistic representation of aerosols and their influence on the cloud phase by acting as INPs is required in models.

*Data availability.* The DARDAR products are available on the Aeris/ICARE data center (https://www.icare.univ-lille.fr/dardar/), (Sourdeval et al., 2018; Delanoë and Hogan, 2010; Ceccaldi et al., 2013). The two CloudSat products 2B-FLXHR-LIDAR_R05 (Henderson et al., 2013; L'Ecuyer et al., 2008) and 2B-CLDCLASS-LIDAR_R05 (Sassen and Wang, 2008) are available on the website of the CloudSat Data Processing Center (DPC) (https://www.cloudsat.cira.colostate.edu/). The Sea Ice Concentration from Nimbus-7 SMMR and DMSP SSM/I-SSMIS Passive Microwave Data Version 1 are available on the website of the National Snow and Ice Data Center (https://nsidc.org/data/nsidc-0051/versions/1) (Cavalieri et al., 1996). The aerosol mixing ratios from the CAMS reanalysis data (Inness et al., 2019) are available on the CAMS Atmosphere Data Store (ADS) website (https://ads.atmosphere.copernicus.eu/cdsapp#!/dataset/cams-global-reanalysis-eac4?tab=overview).

**Appendix A**

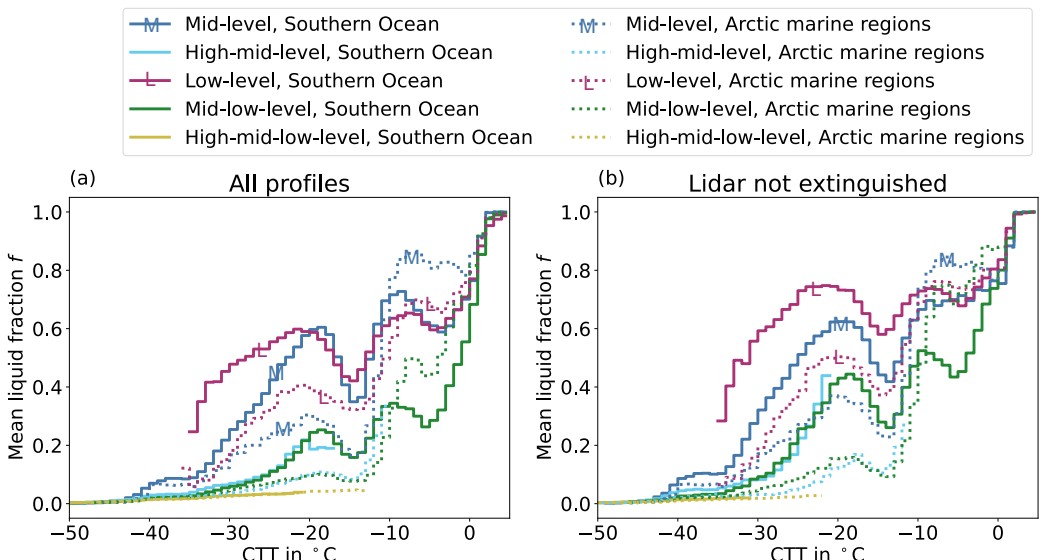

**Figure A1.** Mean liquid fraction of all profiles for different cloud top temperatures (CTT). The liquid fraction is calculated for each vertical cloud column and then averaged for each 1 °C bin of the CTT. The liquid fraction of each profiles is calculated as described in Sec. 3 and Eq. 3. Panel (a) includes all cloud profiles, while panel (b) considers only profiles where the lidar is not fully attenuated.

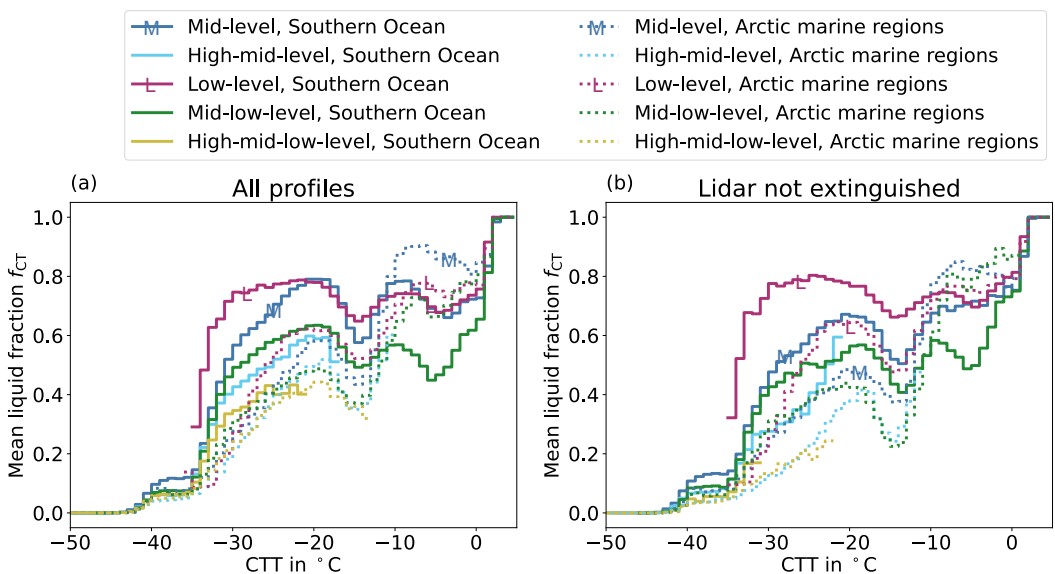

**Figure A2.** Mean liquid fraction based on the phase of all cloud top bins and cloud top temperature (CTT). Panel (a) shows all cloud profiles, while panel (b) considers only profiles where the lidar signal is not fully attenuated.

**Table A1.** Cloud radiative effect summed up over two years and normalised by the total number of observed cloud profiles.

|  | Shortwave | Longwave | Net |
|---|---|---|---|
| Arctic marine regions | -39.1 $\mathrm{Wm^{-2}}$ | 18.7 $\mathrm{Wm^{-2}}$ | -20.4 $\mathrm{Wm^{-2}}$ |
| Southern Ocean | -43.2 $\mathrm{Wm^{-2}}$ | 24.9 $\mathrm{Wm^{-2}}$ | -18.3 $\mathrm{Wm^{-2}}$ |

*Author contributions.* BD performed the data analysis and wrote the initial version of the paper. OS provided the combined dataset of DARDAR-MASK, DARDAR-CLOUD and DARNI, as well as the collocated CAMS aerosol reanalysis data. CH supervised the work and contributed to the conceptualization and design of the study. CH and OS provided feedback on the manuscript. All authors discussed the results and the manuscript and thereby contributed to the writing.

*Competing interests.* CH and OS are members of the editorial board of Atmospheric Chemistry and Physics, and the authors have also no other competing interests to declare.

*Acknowledgements.* This project has received funding from the European Research Council (ERC) under the European Union's Horizon 2020 research and innovation programme under grant agreement No 714062 (ERC Starting Grant "C2Phase") and under grant agreement No 821205 (FORCeS). OS acknowledges funding by the ANR (ANR-20-CE92-0008) for project "CDNC4ACI" and support from CNES through the "EECLAT" project. We thank the ICARE Data and Services Center for providing access to the data used in this study. This paper contains modified Copernicus Atmosphere Monitoring Service information (2019). Neither the European Commission nor ECMWF is responsible for any use that may be made of the information it contains. We would like to thank the CloudSat science team for providing the data products.

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
