# Peer review of "Characterisation of low-base and mid-base clouds and their thermodynamic phase over the Southern Ocean and Arctic marine regions"

_EGUsphere, 2023_

## Author Comment (AC1)

**Response to reviewer comments on "Characterisation of low-base and mid-base clouds and their thermodynamic phase over the Southern and Arctic Ocean" by B. Dietel, O. Sourdeval, C. Hoose**

We thank the anonymous reviewers for their recommendations and comments on the manuscript. Please find the detailed responses below in blue.

**Reply to reviewer 1 comments:**

This paper reports on an extensive study of mid-to-high latitude marine clouds, over ocean and sea ice, on both hemispheres, focusing on cloud phase and potential dependencies on different aspects. The manuscript is fairly well written, the methods is generally well conceived and some of the results are interesting. The paper could be publishable with some – maybe quite a bit – more work, so I'll settle for major revision on this one.

**Main concerns**

My first concern is that the authors are using several standardized datasets based on remote sensing and modeling without showing much insight into the uncertainties in these and how the accumulate uncertainty to the final result. There is a sub-chapter on uncertainty – that comes after the results! This should obviously have come up front at the beginning and be more insightful.

Starting with the primary remote sensing data, it is well understood – and documented – that CloudSat has so-called ground clutter problems, making the lowest several hundred meters unusable – or at least very uncertain. This takes a rather large chunk out of the height interval for the "Low" class clouds here; at least in the Arctic low clouds with a cloud base lower than a few hundred meters tend to dominate. It is also well known that a cloud radar cannot distinguish between cloud layers (or a cloud layer and the surface) if precipitation is falling out of the cloud(s), especially if that is frozen as the radar is completely overwhelmed by the size of ice crystals.

Hence, I wouldn't trust the CBH in these datasets very far and then one can't trust cloud thickness either, except when the lidar detects both top and bottom of the cloud. Therefore, the only way to be sure single-layer clouds are sampled is to only use profiles where the lidar can see the surface; else there is no way to know for sure if the lidar is extinguished or not and if so where. The consequence of this would be a quite strongly biased dataset, with only thin clouds; the subset used here, excluding obvious multi-layer clouds, have already roughly halved the cases (Figure 14; "Other"). So, some – or even many – of the single layer clouds in this study may in fact be two-or-more layer clouds with precipitation in between. And even if the DARDAR data set seems more advanced than the previously used, one must also still remember that it is all a retrieval; it is all dependent on a lot of somewhat ad hoc choices combined with a priori model data, which also has it limitations. This doesn't make this type of data useless; one just has to be ultra-careful and multi-suspicious.

On top of this, all the CRE data are based on calculations, not measurements, and all of the aerosol data is also modeling; probably the best modeling one can get, but it is still surrounded both by uncertainty, errors and other problems because of the way the model is designed and the modeling is set up. Moreover, the modeling is using a limited set of aerosol parameters because of the complexity of the problem, with a lot of uncertain parameterizations, and the availability of computational power. This – again – need not be a show stopper, but it has to be acknowledged and discussed. While I agree that reference to data sources should be as good as references to the models used, with descriptions of the uncertainty in methods, there has to be a discussion here of what this means for this study, and these results.

We moved the section of the uncertainties to before the results and after the description of the methodology and added the following sentences about the uncertainty of ground clutter in L231 ff (revised

version): "Another uncertainty is introduced by ground-clutter of the CloudSat radar signal, but that also highlights the advantage of the usage of several instruments for the cloud phase determination. Nevertheless, the clutter of the radar signal introduces uncertainty. Alexander and Protat (2018) showed that DARDAR underestimates clouds at heights of 0.2 km to 1.0 km by a factor of 3 compared to a surface-based lidar at Cape Grim, Australia from July 2013-February 2014." We also mention the importance of uncertainties for some results like in Line 355-357(preprint), 254-257(preprint), or 283-287(preprint). We also did an analysis only using profiles where the lidar is not fully extinguished for the analysis of the mean liquid fraction as a function of the cloud top temperature, which is shown in the Appendix Fig. A1, Fig. A2 and are described in the section about the uncertainties.

Another problem I is find is the inherent "apples and pears" comparisons between the Southern Ocean, which to a great extent is mid-latitude, and the Arctic which is not. The study of both is important, but I would almost have wished these were two separate papers so the authors didn't fall into the trap of doing this comparison. To start, there is nothing north of 82° N in the Arctic, which is a fairly sizeable fraction of the Arctic Ocean, while there is nothing similar for the Southern Ocean. For example, comparing cases with and without sea north of 60°N with those south of 40 °S is not a fair comparison since there are vast areas of the latter that never see any sea ice at all. Why not limit the latter to south of 60 °S to make a better comparison? Starting at 60°N includes the northern north Atlantic (roughly down to Iceland) with rather particular climate conditions due to the AMOC. And this data was collected over two years, which is not a lot to begin with, but nothing is said about seasonality; for example the mid-winter darkness leaves a much larger foot print when sampling 60-82 °N than when sampling 40-82 °S.

Yes, there are differences due to the different latitude boundaries of the Arctic Ocean and the Southern Ocean, as well as different distributions of land surfaces. One reason of comparing these two regions are large radiative biases in weather and climate models related to clouds and their phase over both of these regions. Both of the regions show high occurrences of clouds, frequent mixed-phase clouds, and clouds existing within the mixed-phase temperature regime, where cloud phase is most uncertain. As this study only investigates clouds over the ocean and excluding clouds over land surfaces, moving the threshold for the northern boundary of the Southern Ocean to 60°S would not lead to the same conditions between the Southern Ocean and the Arctic ocean, as the land surfaces are still differently distributed. Antarctica extends over many longitudes to latitudes around 70°S while there is nothing similar over the Arctic Ocean. Moving the threshold over the Southern Ocean to 60°S would also lead to a strong reduction in the sample size and decrease the statistics. We hope that we could clarify why the investigation and even the comparison of the results of both of these regions can still be useful to improve the understanding of cloud phase and the underlying mechanisms. The following studies (Carlsen and David, 2022; Zhang et al., 2019; Lenaerts et al., 2017) for example also compare clouds over both polar regions. We added the following sentence to the introduction in L61-63 (revised version) : "Zhang et al. (2019) and Lenaerts et al. (2017) compare clouds between both polar regions. Zhang et al. (2019) use ground-based remote sensing instruments while Lenaerts et al. (2017) compare satellite observations with reanalysis and climate simulations." The study of Carlsen and David (2022) is already mentioned in L58-60 and we added "over both hemispheres" to the sentence.

Finally, several multi-panel figures are on the very small side and I have a hard time reading the labeling without magnification. This include Figures 5, 6, 11 & 13, while Figure 10 could have more separation between markers; there is certainly room for it.

We adapted the mentioned figures to improve readability.

**Detailed concerns:**
Lines 63-75: While this is a necessary discussion, this problem is inherent to all remote sensing datasets; as the technology and techniques develops the criteria are shifted. This means progress, but is of course also a problem.

We describe this here, as there is no consistent definition of cloud types in literature, even when using the same dataset based on the same technology and techniques. Furthermore, this should highlight the difficulty of comparing various studies using different definitions for different cloud types. The written lines are intended to embed the used classification which is explained afterwards in the context

of previous and other cloud classifications, which have been used before.

Line 65: Why drag geostationary satellites into this? They have limited or no cover in these regions, and currently there is nothing geostationary for the higher latitudes. Even though it could be done...
Especially for the Southern Ocean there have been several studies investigating clouds with geostationary satellite observations for example with Himawari-8 (Kang et al., 2021) or METEOSAT (Coopman et al., 2021). Even though they don't have the best coverage, useful information has been derived in previous work.

Line 94: What ECMWF product? Operational IFS, ERA5 or something else?
We specified that it is the ECMWF AUXillary (ECMWF-AUX) product provided by ECMWF collocated to CloudSat observations and available with CloudSat products and the DARDAR dataset. We changed the sentence to: "Furthermore, collocated ECMWF AUXillary (ECMWF-AUX) data are used within the retrieval process to categorise the cloud phase (Delanoë and Hogan, 2010). Temperature from this dataset is included in the DARDAR dataset and also used for further analysis. The dataset has a vertical resolution of 60 m and a horizontal resolution of about 1.5 km along the track of the polar-orbiting satellites."

Lines 101-120: Have you seen Arctic sea ice in summer? I would argue it is equally or maybe even more horizontally heterogeneous than the clouds above it! But I agree on the temporal side...
We changed the sentence and added an explanation as to why we think the horizontal resolution is suitable for the purpose of our study. We are interested in the effect of aerosol emitted from the ocean and its influence on cloud phase. The horizontal resolution is suitable, as from heterogeneous ice cover, there would still be an effect of locally emitted aerosol transported over the ice edge, but we are more interested in largely covered sea ice regions and regions with clearly open ocean without sea ice.

Section 2.3: And has it ever been evaluated how good this is in the polar regions?
There are validation reports of the CAMS reanalysis available (`https://atmosphere.copernicus.eu/node/325#fe56bdb4-1bdf-4d47-b46b-261a1ea57243`), but only a validation of total AOD against AERONET stations, as well as comparisons against other reanalysis data are shown. Furthermore, some aerosol types are validated over continental regions like Africa or Europe. We added the following sentence about a comparison of sea salt CAMS reanalysis with station observations in L114-115 (revised version): "Lapere et al. (2023) compare CAMS sea salt reanalysis with a few station observations in the Arctic and Antarctic and show in their Fig. 7 that most stations show strong Pearson correlation coefficients despite partly high normalised mean biases." To our knowledge there are not many validation studies of CAMS reanalysis over polar regions, probably due to the lack of observations for specific aerosol types over polar regions.

Lines 119-123: All of which are uncertain! None of the satellites actually measure any of this first hand.
Indeed, the cloud radiative effect is never measured, as it is always compared to a cloud-free scene with the same conditions otherwise. We rephrased the sentence using the word estimate to clarify that the parameters are not directly measured.

Lines 127-130: A figure with maps of these two areas in polar representation would be useful, to see where different land areas are etc., possibly with lines for maximum and minimum ice extent.
We added a map with the sea ice concentration in March and September for both polar regions (See Fig. 1).

Lines 132-137: And do you trust the provided cloud mask blindly?
No, obviously we don't trust the provided cloud mask blindly. We describe several uncertainties like attenuation of the radar or lidar signal in the interpretation of the results, and how this could influence the result, for example in Lines 254-257(preprint version)/296-301(revised version), or 283-287(preprint version)/327-331 (revised version). Furthermore, we also have a section about uncertainties with Lines 533-541(preprint version)/223-231 (revised version) describing uncertainties of the cloud phase data.

As already mentioned above, we added the following sentences about the uncertainty of ground clutter in L231 ff (revised version): "Another uncertainty is introduced by ground-clutter of the CloudSat radar signal, but that also highlights the advantage of the usage of several instruments for the cloud phase determination. Nevertheless, the clutter of the radar signal introduces uncertainty. Alexander and Protat (2018) showed that DARDAR underestimates clouds at heights of 0.2 km to 1.0 km by a factor of 3 compared to a surface-based lidar at Cape Grim, Australia from July 2013-February 2014."

Line 142: Curious why you selected 2 km as the upper limit, and also what the lower limit was set to. As stated in Line 145-146 (preprint version)/150-151 (revised version) these thresholds are based on the definitions from the World Meteorological Organization (2017), which provides a cloud atlas for cloud type definitions with specific heights. We don't use an explicit lower limit and added 0 km as a threshold to L155 and L157 (revised version). The two figures below show 2-dimensional histograms of CBH and CTH for single-layer clouds (Fig. R1) and low-level clouds (Fig. R2).

[Figure]

Figure R1: 2-dimensional histograms of the cloud base height and cloud top height of single-layer clouds.

[Figure]

Figure R2: 2-dimensional histograms of the cloud base height and cloud top height of low-level clouds.

Equation (2): Is this really necessary?
As the equation describes the parameter which is used for the classification of different clouds types,

we would argue that it is necessary to write the exact definition of these thresholds. This can also be useful to ensure the reproducibility and possible comparisons in future.

Equation (3) & (4): So, you just assume that half of the mixed-phase pixels are liquid? Is there any basis for this, and why not 0.4 or 0.6? Or is it a "matter of convenience" lacking a better solution? It is an ad-hoc assumption. The mixed-phase pixel classification is mainly based on both a radar and a lidar signal from the same height hinting at the presence of both ice particles and liquid droplets within the same pixel. As we don't have more information about the actual fraction, we assume that half of the pixel is liquid and half of the pixel consists of ice particles. We added the following sentence to the manuscript: "Due to the absence of better information it is assumed that half of each mixed-phase pixel consists of liquid droplets and half of it consists of frozen ice crystals, as the mixed-phase category is mainly based on a signal from both the radar and the lidar."

Lines 168-172: What are the intervals? The interval refer to different cloud types. Here we compare how many profiles of a specific cloud type occur over open ocean and sea ice. To clarify this, we added the following part "... with the interval referring to the fraction for different cloud types."

Line 185: "... satellite track."? We changed it as suggested.

Line 187: Heterogenous variability in a single cloud layer on scales > 1km could still be the same cloud, especially if it is less than solid. I have no other suggestion; you're probably erring on the safe side here and the scales could be larger. Yes, there can be discontinuities within a cloud, which would be considered as two separate clouds, but there is always the question what can still be considered as being the same cloud? We have also done some sensitivity tests using a time threshold of 0.4 s and 0.6 s, which are not shown. This would mean that cloud profiles with one or two profiles missing in between would still be considered as the same cloud. As expected the mean horizontal extent increases for larger time thresholds. We can see the maximal increase by 69 % for 0.4 s and by 98 % for 0.6 s compared to the results using a threshold of 0.2 s. The vertical extent is similar and the changes are in the range of an 18 % decrease and a 7 %increase.

Lines 198-207 and elswhere: Please be a bit more imaginative when describing a figure; I can see myself what it shows, so I want the interpretation; not a repetition of what I can see. We slightly changed the wording of the paragraph 242-248 (revised version). We shortened the factual description and have put more weight on the interpretation.

213-216: Do not compare apples and pears! We compare two regions showing large biases in the representation of clouds and their phase in models. We investigate and compare the properties of clouds in these regions to improve the understanding of the underlying processes and their representation in models in future. Despite the differences in land fraction, latitude etc., there are remarkable similarities in the cloud properties, which we find worth mentioning.

Line 226-237: When comparing to another study, it is essentiall to know where and for what time period that study was done. Maybe differences are to be expected, if the seasons or regions are very different. We changed the sentence to: "Sassen and Wang (2008) investigated the frequency of specific cloud types as a function of the latitude with the 2B-CLDCLASS dataset based on global CloudSat observations from 15 June 2006 to 15 June 2007 distinguishing between clouds over ocean and clouds over land." This should clarify the region and time of the study, but the time and the region is probably not the reason for the differences. One reason can be that we include observations of the lidar, and another reason is probably related to the differences in the cloud type definitions as stated in the Lines 233-235 (preprint)/265-267(revised version).

Line 243-245: While it seems intuitive that liquid fraction would increase if you use only profiles that are not fully extinguished, and also that mixed-phase fraction must decrease if also ice fraction is increasing, the latter is not obvious; could you elaborate.

The explanation for these changes is most likely that the lidar is mostly extinguished in mixed-phase clouds often showing a liquid layer at cloud top and precipitating ice below. This reduces the number of mixed-phase clouds if we only consider profiles, where the lidar is no fully extinguished and leads to an increase of the relative frequency of liquid and ice clouds. The figure below shows the decrease of the number of profiles if only profiles are considered where the lidar signal is not fully attenuated. The absolute numbers of liquid, mixed-phase, and ice cloud profiles are reduced (see Fig. R3), but the strongest decrease can be seen in mixed-phase clouds, which is why this also shows a decrease in the relative frequency, while the relative frequency of liquid and ice clouds correspondingly increases in most cases. We also changed some sentences in this paragraph L296-301 (revised version) to clarify this: "Regarding the differences in the results considering all profiles of a cloud type, shown in the outer pie charts in Fig. 6 and only considering profiles where the lidar is not fully attenuated, we can generally see a decrease of the fraction of mixed-phase profiles, which indicates that in mixed-phase profiles the lidar is frequently attenuated. The reduction of the absolute numbers of profiles is strongest for mixed-phase clouds, which is the reason for the decrease of the relative fraction of mixed-phase profiles, while the relative fraction of ice and liquid profiles mostly increases. This might lead to uncertainties in the further analysis. "

[Figure]

Figure R3: Fraction of the number of cloud type profiles, where the lidar signal is fully attenuated.

Figure 4 and related text: To a first order, this only shows that temperature decreases upwards in the atmosphere; rather trivial don't you think?

Yes, to a first order of course temperature decreases with height. Nevertheless, you can see differences between the Southern Ocean and the Arctic Ocean in the frequency of different cloud types occurring at different heights. The purpose of this analysis was mainly to investigate which cloud types occur most frequent in the mixed-phase temperature regime, where cloud phase is most uncertain. We use cloud top temperature, because usually the ice formation mostly occurs at cloud top, where temperature is coldest within the vertical cloud column.

Figure 5: Why is HML lacking liquid when M has plenty? Is that because the lidar can't see through more than the top of juicy deep clouds, and there is really some liquid farther down that is missed?

Liquid refers to completely liquid clouds in Fig. 5. One reason is for example shown in Fig. 4. If you compare CTT of M and HML, you can see that most HML have cloud top temperatures below -38°C which explains, that most of HML clouds include frozen ice crystals and are not completely liquid, while most M clouds have CTT larger than -38°C and can therefore stay liquid. But yes there is also an uncertainty related to the limited penetration depth of the lidar, where we don't know if there is more liquid in lower parts of the cloud. But for HML clouds this would then also be mixed-phase clouds

as they mostly have ice in the upper part of the cloud. We added an explanation to the description that the lack of liquid clouds is probably related to low cloud top temperatures and homogeneous ice formation.

Lines 268-269: Isn't this rather trivial? The vertical structure is such that there is simple much more vertical distance with sub-freezing temperatures than with temperatures above zero. Hence the likelihood of finding a really deep ice cloud must much larger than to find a similarly deep liquid cloud. While it is expected that ice clouds have a larger vertical extent for the reason given by the reviewer, the larger horizontal extent of liquid clouds hints to preferred cloud structure and morphology depending on the phase.

Lines 281-293: Could it be that the lidar is just extinguished and then you can't know how thick the liquid layer really is? While for mixed-phase clouds you have use of the radar even when the lidar is extinguished?
Yes that is true and already stated in the Lines 283-287(preprint)/327-331(revised version) "Therefore, the extinction of the lidar signal, due to strong attenuation, probably has a strong influence on the result of the small vertical extent of the liquid clouds. Thus, this results is at least partly coming from the limited penetration depth of the lidar signal in supercooled liquid clouds. Nevertheless, there have been studies using other observation techniques, such as ground based remote sensing, showing that supercooled liquid cloud layers tend to be shallow (Ansmann et al., 2009)."

Figure 6: Way to small; make two figures – and please recalculate everything to precentages instead of "# of clouds".
We changed the figure and show now the percentages instead of absolute numbers of cloud profiles. We also changed the unit of the horizontal extent to km assuming a horizontal distance of 1.1 km for one profile. We keep the figure as one figure, because we think it simplifies the comparison between the different panels of the 2-dimensional histograms, as well as the mean values of the horizontal and the vertical extent.

Figure 7 and text on the "dips": I find this intriguing and at the same time I do not find the suggested explanation very credible. I can buy that there are regime shifts as temperature changes, but that there would be a band of temperatures where something particular would happen while both higher and lower temperatures would be similar. At the same time, the results for the north and the south are sufficiently similar to make this either common (global) processes or – and this would be my guess – a consequence of something in the retrieval process. The latter would at least have to be excluded before trying to "fit a round object into a square hole".
In general, the occurrence of specific processes can be very dependent on the temperature. Libbrecht (2005); Fukuta and Takahashi (1999) or Chen and Lamb (1994) have for example shown that the shape of ice particles strongly depends on specific temperatures, and there are also regimes with similar shapes. Also, the occurrence of specific secondary ice processes is know to occur at specific temperature ranges. To our knowledge, the retrieval process can not explain the dips. Furthermore, there have been other studies showing these dips based on different retrievals using only CloudSat observations (Riley and Mapes, 2009), using only Calipso observations (Nagao and Suzuki, 2022), using both CloudSat and Calipso but a different retrieval (Zhang et al., 2014), but also in ground-based remote sensing observations (Zhang et al., 2019). The presence of these "dips" in various studies using different observations and retrievals makes us quite confident, that it is not only retrieval based. Surely, it has to be investigated further and the provided explanation is only a hypothesis so far, as clearly indicated in the text.

Line 340-341: The models do what they are told to do; this is all parameterized in the models and if we can't explain these dips, why would you expect the models to be able to?
Models can reveal interdependencies and joint effects of the implemented processes. If some processes are for example driven by dynamics or other microphysical processes, we would assume to see signals, without telling the model explicitly to show a specific behaviour.

Lines 350-351: I think this would be contrary to many surface-based remote sensing studies, as well as a few aircraft profiles. Maybe you have included ground clutter here?

Yes, as described in lines 355-357(preprint)/399-401(revised version), it is not yet clear if this is related to uncertainties. We added ground clutter as one of the possible reasons in line 357. "... uncertainties like ground clutter"

Section 4.2.4.: Like mentioned earlier I would like this to be more comparable, by restricting the are to the ocean where there is some ice possible.
This would strongly reduce the number of observations, especially if we only consider the region, where then both is possible open ocean and sea ice.

Lines 380-381: I'm no expert here, but sea-salt I get is an excellent CCN. But why do you think it is also a good INP? I'd like to see that argument.
Sea Spray aerosols often contain biological components from the sea surface microlayer which are highly ice active at high temperatures. This has been shown in many previous studies by Porter et al. (2022); Inoue et al. (2021); Twohy et al. (2021); Ickes et al. (2020); Wilson et al. (2015). We slightly changed the sentence in lines 384-386 to: "Sea salt acts hereby as a proxy for sea spray aerosols, as the ice nucleating part of the aerosols are usually biological components from the sea surface microlayer, like microorganisms acting as INP at high temperatures (Burrows et al., 2013; Després et al., 2012)."

Lines 407-408: Because you can't see the liquid in thick deep clouds doesn't mean it's not there!
We changed the sentence in line 407-408(preprint)/451-452(revised version) to: "High-mid-low-level clouds show no signal, possibly because they mostly consist of ice (see Fig. 7), but the attenuation of the remote sensing signals may introduce uncertainties here."

Line 412: What is the rationale for this assumption? Is it some insights into the modeling of sea-salt aerosols, or is it just that it would fit well with something you like to believe? I suggest you look into what the model really does and what the source term(s) for sea-salt aerosol is.
The rationale for this assumption is that many studies have shown that sea spray aerosols frequently include biological components which are highly ice active (Porter et al., 2022; Inoue et al., 2021; Twohy et al., 2021; Ickes et al., 2020; Wilson et al., 2015). The production mechanism of sea salt and sea spray is similar, and mainly depends on wind velocities (which is represented in the model) leading to the air bubbles bursting at the sea surface. Therefore, we can assume that the model output of sea salt can be used as a proxy for sea spray aerosol containing other chemical components than salt like organics.

Line 426-435: Drop this whole paragraph; it is irrelevant in the context of this work. The satellite data has no way of distinguishing between coupled and uncoupled systems and CAMS in fundamentally incapable of simulating it, and hence any effects it would or could have on the aerosol contribution. Youre grasping for straws here.
We don't think that this is irrelevant in the context of this work, as the coupling of clouds can be connected to the presence of marine organic INPs and thereby influence cloud phase, which is exactly the topic of this work. To clarify this, we changed the first sentence of the paragraph (L471 revised version) to: "Additional support for the hypothesis that sea spray aerosol strongly impact the phase of polar low clouds is given by the work of Griesche et al. (2021), who found a dependence of the Arctic cloud phase on the surface coupling, connected with marine INPs, but only for temperatures warmer than -15 °C, while we see a correlation of sea salt aerosol with the cloud phase for much lower temperatures."

Figure 13: How are these results averaged? What about sea ice or not, and what about the seasonal winter darkness, when shortwave CRE does not exist by definition.
We added a sentence at the beginning of the section to clarify that both clouds over sea ice and clouds over ocean are considered, only clouds over land surfaces are excluded, as for the whole study. We also investigated the differences in CRE over sea ice and ocean, but didn't include this in the study. All cloud profiles over the 2-year period are considered as for the rest of the study, which is why we don't write this again explicitly. Yes, during the polar night, no short-wave CRE is defined, and nan values are provided by the 2B-FLXHR-LIDAR dataset and thereby not considered for the average.

Line 563: As for vertical thickness and cloud phase, the only thing you have shown really is that temperature decreases with height in the atmosphere. More low temperatures, more ice. And in the

horizontal it is not thickness; it is distance.

First, yes temperature decreases with height, but as already mentioned earlier the possibility of liquid clouds can also reach quite large heights including the mixed-phase temperature regime with supercooled liquid water. We changed the wording to horizontal extent.

Lines 567-570: I would be careful here; it might also be an artefact of the method. The fact that the Southern Ocean and the Arctic are so similar despite having very different aerosol climates suggests that it is not a microphysical process; rather something else artificial.

We agree with the reviewer that the origin of this feature has to be further investigated in future studies. If the reason for these dips is for example the dendritic growth of ice particles and thereby induced secondary ice, this is not strongly correlated with the aerosol climates, but is more dependant on temperature and humidity. Therefore, we clearly state that these are possible reasons which need to be investigated in further studies. As explained above (see answer to the question about Fig. 7 and "dips"), we also cite several other studies, which have seen a similar signals using different methodologies and different retrievals, which makes us confident that the signal is not only based on our specific method and the specific retrieval.

**References**

S. P. Alexander and A. Protat. Cloud Properties Observed From the Surface and by Satellite at the Northern Edge of the Southern Ocean. *Journal of Geophysical Research: Atmospheres*, 123(1): 443–456, 2018. ISSN 2169-8996. doi: 10.1002/2017JD026552. https://agupubs.onlinelibrary.wiley.com/doi/abs/10.1002/2017JD026552.

Tim Carlsen and Robert O. David. Spaceborne Evidence That Ice-Nucleating Particles Influence High-Latitude Cloud Phase. *Geophysical Research Letters*, 49(14):e2022GL098041, 2022. ISSN 1944-8007. doi: 10.1029/2022GL098041. https://onlinelibrary.wiley.com/doi/abs/10.1029/2022GL098041.

Jen-Ping Chen and Dennis Lamb. The Theoretical Basis for the Parameterization of Ice Crystal Habits: Growth by Vapor Deposition. *Journal of the Atmospheric Sciences*, 51(9):1206–1222, May 1994. ISSN 0022-4928, 1520-0469. doi: 10.1175/1520-0469(1994)051⟨1206:TTBFTP⟩2.0.CO;2. https://journals.ametsoc.org/view/journals/atsc/51/9/1520-0469_1994_051_1206_ttbftp_2_0_co_2.xml.

Q. Coopman, C. Hoose, and M. Stengel. Analyzing the Thermodynamic Phase Partitioning of Mixed Phase Clouds Over the Southern Ocean Using Passive Satellite Observations. *Geophysical Research Letters*, 48(7):e2021GL093225, 2021. ISSN 1944-8007. doi: 10.1029/2021GL093225. https://onlinelibrary.wiley.com/doi/abs/10.1029/2021GL093225.

Julien Delanoë and Robin J. Hogan. Combined CloudSat-CALIPSO-MODIS retrievals of the properties of ice clouds. *Journal of Geophysical Research: Atmospheres*, 115(D4), 2010. ISSN 2156-2202. doi: 10.1029/2009JD012346. https://agupubs.onlinelibrary.wiley.com/doi/abs/10.1029/2009JD012346.

Norihiko Fukuta and Tsuneya Takahashi. The Growth of Atmospheric Ice Crystals: A Summary of Findings in Vertical Supercooled Cloud Tunnel Studies. *Journal of the Atmospheric Sciences*, 56 (12):1963–1979, June 1999. ISSN 0022-4928, 1520-0469. doi: 10.1175/1520-0469(1999)056⟨1963:TGOAIC⟩2.0.CO;2. https://journals.ametsoc.org/view/journals/atsc/56/12/1520-0469_1999_056_1963_tgoaic_2.0.co_2.xml.

Hannes J. Griesche, Kevin Ohneiser, Patric Seifert, Martin Radenz, Ronny Engelmann, and Albert Ansmann. Contrasting ice formation in Arctic clouds: Surface-coupled vs. surface-decoupled clouds. *Atmospheric Chemistry and Physics*, 21(13):10357–10374, July 2021. ISSN 1680-7316. doi: 10.5194/acp-21-10357-2021. https://acp.copernicus.org/articles/21/10357/2021/.

Luisa Ickes, Grace C. E. Porter, Robert Wagner, Michael P. Adams, Sascha Bierbauer, Allan K. Bertram, Merete Bilde, Sigurd Christiansen, Annica M. L. Ekman, Elena Gorokhova, Kristina

Höhler, Alexei A. Kiselev, Caroline Leck, Ottmar Möhler, Benjamin J. Murray, Thea Schiebel, Romy Ullrich, and Matthew E. Salter. The ice-nucleating activity of Arctic sea surface microlayer samples and marine algal cultures. *Atmospheric Chemistry and Physics*, 20(18):11089–11117, September 2020. ISSN 1680-7316. doi: 10.5194/acp-20-11089-2020. https://acp.copernicus.org/articles/20/11089/2020/.

Jun Inoue, Yutaka Tobo, Fumikazu Taketani, and Kazutoshi Sato. Oceanic Supply of Ice-Nucleating Particles and Its Effect on Ice Cloud Formation: A Case Study in the Arctic Ocean During a Cold-Air Outbreak in Early Winter. *Geophysical Research Letters*, 48(16):e2021GL094646, 2021. ISSN 1944-8007. doi: 10.1029/2021GL094646. https://onlinelibrary.wiley.com/doi/abs/10.1029/2021GL094646.

Litai Kang, Roger Marchand, and William Smith. Evaluation of MODIS and Himawari-8 Low Clouds Retrievals Over the Southern Ocean With In Situ Measurements From the SOCRATES Campaign. *Earth and Space Science*, 8(3):e2020EA001397, 2021. ISSN 2333-5084. doi: 10.1029/2020EA001397. https://onlinelibrary.wiley.com/doi/abs/10.1029/2020EA001397.

Rémy Lapere, Jennie L. Thomas, Louis Marelle, Annica M. L. Ekman, Markus M. Frey, Marianne Tronstad Lund, Risto Makkonen, Ananth Ranjithkumar, Matthew E. Salter, Bjørn Hallvard Samset, Michael Schulz, Larisa Sogacheva, Xin Yang, and Paul Zieger. The Representation of Sea Salt Aerosols and Their Role in Polar Climate Within CMIP6. *Journal of Geophysical Research: Atmospheres*, 128(6):e2022JD038235, 2023. ISSN 2169-8996. doi: 10.1029/2022JD038235. https://onlinelibrary.wiley.com/doi/abs/10.1029/2022JD038235.

Jan T. M. Lenaerts, Kristof Van Tricht, Stef Lhermitte, and Tristan S. L'Ecuyer. Polar clouds and radiation in satellite observations, reanalyses, and climate models. *Geophysical Research Letters*, 44(7):3355–3364, 2017. ISSN 1944-8007. doi: 10.1002/2016GL072242. https://onlinelibrary.wiley.com/doi/abs/10.1002/2016GL072242.

Kenneth G Libbrecht. The physics of snow crystals. *Reports on Progress in Physics*, 68(4):855–895, April 2005. ISSN 0034-4885, 1361-6633. doi: 10.1088/0034-4885/68/4/R03. https://iopscience.iop.org/article/10.1088/0034-4885/68/4/R03.

Takashi M. Nagao and Kentaroh Suzuki. Characterizing Vertical Stratification of the Cloud Thermodynamic Phase With a Combined Use of CALIPSO Lidar and MODIS SWIR Measurements. *Journal of Geophysical Research: Atmospheres*, 127(21):e2022JD036826, 2022. ISSN 2169-8996. doi: 10.1029/2022JD036826. https://onlinelibrary.wiley.com/doi/abs/10.1029/2022JD036826.

Grace C. E. Porter, Michael P. Adams, Ian M. Brooks, Luisa Ickes, Linn Karlsson, Caroline Leck, Matthew E. Salter, Julia Schmale, Karolina Siegel, Sebastien N. F. Sikora, Mark D. Tarn, Jutta Vüllers, Heini Wernli, Paul Zieger, Julika Zinke, and Benjamin J. Murray. Highly Active Ice-Nucleating Particles at the Summer North Pole. *Journal of Geophysical Research: Atmospheres*, 127(6):e2021JD036059, 2022. ISSN 2169-8996. doi: 10.1029/2021JD036059. https://onlinelibrary.wiley.com/doi/abs/10.1029/2021JD036059.

Emily M. Riley and Brian E. Mapes. Unexpected peak near -15°C in CloudSat echo top climatology. *Geophysical Research Letters*, 36(9), 2009. ISSN 1944-8007. doi: 10.1029/2009GL037558. https://onlinelibrary.wiley.com/doi/abs/10.1029/2009GL037558.

Cynthia H. Twohy, Paul J. DeMott, Lynn M. Russell, Darin W. Toohey, Bryan Rainwater, Roy Geiss, Kevin J. Sanchez, Savannah Lewis, Gregory C. Roberts, Ruhi S. Humphries, Christina S. McCluskey, Kathryn A. Moore, Paul W. Selleck, Melita D. Keywood, Jason P. Ward, and Ian M. McRobert. Cloud-Nucleating Particles Over the Southern Ocean in a Changing Climate. *Earth's Future*, 9(3):e2020EF001673, 2021. ISSN 2328-4277. doi: 10.1029/2020EF001673. https://onlinelibrary.wiley.com/doi/abs/10.1029/2020EF001673.

Theodore W. Wilson, Luis A. Ladino, Peter A. Alpert, Mark N. Breckels, Ian M. Brooks, Jo Browse, Susannah M. Burrows, Kenneth S. Carslaw, J. Alex Huffman, Christopher Judd, Wendy P. Kilthau, Ryan H. Mason, Gordon McFiggans, Lisa A. Miller, Juan J. Nájera, Elena Polishchuk, Stuart Rae, Corinne L. Schiller, Meng Si, Jesús Vergara Temprado, Thomas F. Whale, Jenny P. S. Wong,

Oliver Wurl, Jacqueline D. Yakobi-Hancock, Jonathan P. D. Abbatt, Josephine Y. Aller, Allan K. Bertram, Daniel A. Knopf, and Benjamin J. Murray. A marine biogenic source of atmospheric ice-nucleating particles. *Nature*, 525(7568):234–238, September 2015. ISSN 0028-0836, 1476-4687. doi: 10.1038/nature14986. http://www.nature.com/articles/nature14986.

World Meteorological Organization. Definitions of clouds. https://cloudatlas.wmo.int/clouds-definitions.html, 2017.

Damao Zhang, Tao Luo, Dong Liu, and Zhien Wang. Spatial scales of altocumulus clouds observed with collocated CALIPSO and CloudSat measurements. *Atmospheric Research*, 149:58–69, November 2014. ISSN 0169-8095. doi: 10.1016/j.atmosres.2014.05.023. http://www.sciencedirect.com/science/article/pii/S0169809514002324.

Damao Zhang, Andrew Vogelmann, Pavlos Kollias, Edward Luke, Fan Yang, Dan Lubin, and Zhien Wang. Comparison of Antarctic and Arctic Single-Layer Stratiform Mixed-Phase Cloud Properties Using Ground-Based Remote Sensing Measurements. *Journal of Geophysical Research: Atmospheres*, 124(17-18):10186–10204, 2019. ISSN 2169-8996. doi: 10.1029/2019JD030673. https://onlinelibrary.wiley.com/doi/abs/10.1029/2019JD030673.

---

## Author Comment (AC2)

**Response to reviewer comments on "Characterisation of low-base and mid-base clouds and their thermodynamic phase over the Southern and Arctic Ocean" by B. Dietel, O. Sourdeval, C. Hoose**

We thank the anonymous reviewer for their recommendations and comments on the manuscript. Please find the detailed responses below in blue.

**Reply to reviewer 2 comments:**

This study presents a thorough characterization of low-base and mid-base clouds in the Polar regions, including their cloud phase and cloud radiative effect (CRE) decomposed by different cloud types. The authors use 2 years of satellite data (CloudSat and CALIPSO) to investigate the influence of aerosols, surface type (ocean/sea ice) and cloud type on the cloud phase and CRE, respectively. This characterization based on observational data provides an important tool to validate model simulations in the future, as models currently struggle to simulate cloud phase in the high latitudes correctly. The study uses a consistent definition of cloud types, which is very important based on the presented results and needs to be considered when comparing to other studies.

The manuscript is extremely well structured and is well written, so it is easy for the reader to follow the storyline. The methods are clear and described in a concise way. The figures are generally of very good quality. Some figures display rather complex content to decompose the relationships with respect to the different cloud types. However, the authors made a good job in choosing different ways to visualize this. I would like to mention that I find the introduction and the discussion of the results with respect to other studies outstanding. This makes it much easier to put the findings into context. I do not have any major comments, but please find some specific comments and technical corrections below.

Specific comments

Line 24: Could you specify 'complex microphysics' a bit more?

We slightly changed the sentence in L24 to: "Models with more complex microphysics than only temperature dependent liquid and ice partitioning tend to show a better representation of the liquid phase fraction, but all models struggle to generate the correct shortwave reflection south of 55 °S (Cesana et al., 2022)." The more complex microphysics refers to the partitioning between liquid and ice, which is not just based on temperature as it is for simple microphysics schemes. More complex microphysics include more processes influencing this partitioning. In the Supplementary Information of the cited paper of Cesana et al. (2022), Table S1 shows an overview of the different CMIP5 models, and if they use only a temperature dependent liquid/ice partitioning or a more complex one, where further literature for each model is also provided.

L44: specify where this underestimation occurs – at the surface?
Yes, we added "at the surface" at the end of the sentence.

L124: It would be helpful to mention that you are investigating the CRE at the TOA already earlier in the paragraph (closer to the equation).
Yes, we changed the first sentence of the paragraph to make it clear that we only focus on the CRE at TOA. As we changed the first sentence to "We analyse the cloud radiative effect (CRE) of various cloud types at the top of the atmosphere (TOA) using the version ...", we then remove the last sentence of the paragraph (only stating that we investigate CRE at TOA).

L144: I think you could build a bit upon how you include Fig. 2 in the manuscript. This seems to me like a very nice justification of the introduction of Zmax for cloud classification based on atmospheric temperatures, however, so far you only introduce Fig. 2 with regards to the dashed lines. Please elaborate a bit here.

We added the following sentence in L151(revised version), to connect the threshold Zmax to the temperature distribution shown in Fig. 2. "Regarding the vertical distribution of the annual mean temperatures in Fig. 2, the threshold Zmax is also mostly parallel to the isotherms, which shows one of the reasons for the chosen threshold decreasing polewards. Furthermore, the threshold Zmax is in the upper part of the mixed-phase temperature regime."

L148: You are giving estimates about how often different cloud types occur over the different surface types, and also that you reduce the number of profiles by 50% by excluding multi-layer cloud scenes. Another helpful estimate would be how many clouds actually have CBH between the lower CloudSat detection limit of about 1 km and your lower threshold of 2 km? What is the exact reason that you are limiting yourself to clouds with CBH above 2 km instead of 1 km?
The threshold of 2 km is based on the definitions of the World Meteorological Organisation for low clouds, which is why we use this threshold for distinguishing cloud types. But we also include clouds with CBH lower than 2km, as illustrated in Fig. 1. The two figures below show 2-dimensional histograms of cloud base heights and cloud top heights for single-layer clouds (Fig. R1) and low-level clouds (Fig. R2).

[Figure]

Figure R1: 2-dimensional histograms of the cloud base height and cloud top height of single-layer clouds.

[Figure]

Figure R2: 2-dimensional histograms of the cloud base height and cloud top height of low-level clouds.

L167: What are the ranges in occurrences referring to? It seems like the calculation of fraction of cloudy profiles over e.g. open ocean would yield one value only.

Yes, the ranges of values refers to the different cloud types. We changed the sentence (L179-181 revised version) to make that more clear: "Over the Southern Ocean 72 % to 82 % of the cloud profiles occur over open ocean, 10 % to 16 % of the cloud profiles are over closed sea ice with the interval referring to the fraction for different cloud types."

Methodology: I suggest clarifying/summarizing at the beginning the two different approaches: (1) cloud object analysis of horizontally connected clouds, (2) individual cloud profile analysis. Please also specify that you are calculating the statistics over all profiles that are e.g. over open ocean, and that no spatial analysis is done. The use of the word 'pixel' in the vertical dimension for the liquid fraction calculation is a bit misleading, at least for me. I relate 'pixel' to something in the horizontal dimension. Maybe you could instead use vertical bin to clearly distinguish the two dimensions?

We added the following two sentences about the two approaches after the description of the cloud type classification: "We generally use two approaches, (1) the statistical analysis of individual cloud profiles, and (2) the analysis of cloud objects, which are horizontally connected cloud profiles. In most parts of the paper approach (1) is used, while in Sec. 5.2.1 approach (2) is used." We changed the word pixel to vertical bin or cloud top bin. We also clarified in L195-196, that for the comparison between ocean-sea ice and low-high aerosol concentration no spatial analysis is done: "These analysis are based on the statistics of individual profiles and no spatial analysis is done for this part." Most parts of the paper are based on the statistical analysis of individual cloud profiles, and only Sec. 5.2.1 (4.2.1 in preprint) is based on connected cloud profiles.

L221: Could you give numbers for that? Are these clouds more frequent in the Arctic with regard to the relative frequency, or in absolute numbers? (as the overall relative frequency of low-level clouds in the Southern Ocean is higher).

It is with regard to the relative frequency with respect to all single-layer clouds, which is shown in Fig. 4 (5 in revised version). It is not in absolute numbers. We add below a figure (Fig. R3) showing the absolute numbers of cloud type profiles as a function of the cloud top temperatures. We also add below a different visualization of Fig. 4 in the paper (Fig. R4) to clearly show the differences. Nevertheless, we think that the figure with stacked frequencies (shown in the preprint) shows a nicer overview of the most frequent cloud types in the mixed-phase temperature regime with better readability. We slightly changed the sentence in L221 (L274-275 revised version) to make it more clear that it refers to the relative frequency and not absolute numbers of occurrence: "The relative frequency of low-level clouds is slightly higher over the Arctic Ocean compared to the Southern Ocean for CTT colder than -13 °C."

[Figure]

Figure R3: Histogram of the absolute numbers of observed cloud types in two years with respect to their cloud top temperature (CTT).

[Figure]

Figure R4: Relative fraction (not stacked) of different cloud types with respect to single-layer clouds as a function of cloud top temperatures (CTTs). The black line shows the lower temperature boundary of the mixed-phase temperature regime at -38 °C.

L227: The paragraph where you compare the cloud type occurrences to Sassen and Wang (2008) seems very relevant, however it seems like you are comparing it to your frequency values based on Fig. 3. Moving this paragraph before you start discussing the dependence on CTT (Fig. 4) would make a bit more sense to me.
Yes, that is a good point and we changed the order and moved the paragraph with Sassen and Wang before the discussion of CTT dependence.

L254: I find it a bit counter-intuitive that you get more low-level liquid clouds if you only consider profiles where the lidar is not fully attenuated as compared to all cloudy profiles. Your 'phase flag' approach basically assigns a liquid/mixed/ice flag based on your calculated liquid fraction. This should not lead to a higher frequency of liquid clouds because you can detect more liquid (as you say on L243), as your phase flag is not necessarily sensitive to the total amount of liquid water. The lidar gets mainly attenuated by the liquid part of the profile, so most often you should have the case where you miss ice in the lower parts of the cloud as the lidar is already attenuated by liquid above (e.g., in the very abundant case of liquid-topped mixed-phase clouds in the high-latitudes). Is this due to the fact that these are only relative frequencies here, and due to the larger vertical extent of mixed-phase clouds you actually filter out more mixed-phase clouds than liquid-only clouds when accounting for only not fully attenuated profiles?

[Figure]

Figure R5: Fraction of the number of cloud type profiles, where the lidar signal is fully attenuated.

The increase of the fraction of liquid clouds is mainly based on the relative fraction and not on an increased detection of the liquid phase. We changed some sentences in this paragraph (also L296 ff.). Indeed, accounting for only not fully attenuated profiles reduces the absolute number of all liquid, mixed-phase, and ice clouds (see Fig. R5), but the strongest decrease can be seen in mixed-phase clouds, which is why this also shows a decrease in the relative frequency, while the relative frequency of liquid and ice clouds correspondingly increases.

L295: Could you state the extensions (horizontal/vertical) from your analysis here as well to compare the values? As the horizontal extent is given in number of profiles and not in km.
We added the numbers of the horizontal extent of mid-level clouds which are between 3.89 km to 6.24 km. We also changed the unit of the horizontal extent in the figure from number of profiles to km assuming a horizontal distance of 1.1 km.

L335: Very nice job in putting the results into perspective to other studies. Regarding one point you are mentioning briefly I was wondering whether you have looked into Fig. 7 on a seasonal basis? I am wondering whether sea ice leads to the fact that supercooled liquid clouds can get maintained at much colder temperatures without the open ocean as a potential source of INPs? A seasonal look into this very interesting data set could potentially disentangle some of the reasons the authors mention.
We did an analysis for another study for two seasons shown in Fig. R6. We saw an expected different coverage of cloud top temperatures in the different seasons, but the local minima are visible in both seasons. The argument that sea ice leads to the maintained supercooled liquid water at cold cloud top temperatures due to less INPs is discussed in Chapter 4.2.4, where we directly analyze cloud phase as a function of sea ice coverage and analyze aerosol reanalysis data.

[Figure]

Figure R6: Mean liquid fraction of the period from 11 August - 09 September in panel (a) and of the period from 31 January - 28/29 February in panel (b) both of the years 2007 and 2008.

L404: In addition, as Papakonstantinou-Presvelou et al. (2022) investigate ice-only clouds, they might look into former mixed-phase clouds at a different stage in their lifecycle. These might already be completely glaciated clouds, so a comparison of the ice crystal number with the liquid fraction presented in this study is not easy.

We agree. We already state in L399-400 (L444 revised version) that "[...] Papakonstantinou-Presvelou et al. (2022) investigate only ice clouds, while our study investigates the general cloud phase." When describing their results, we also clearly state that they investigate ice crystal number concentrations, while we investigate cloud phase/liquid fraction. Although a direct comparison is not easy, we still think that the result of Papakonstantinou-Presvelou et al. (2022) should be part of the discussion as both studies investigate differences of low cloud properties over sea ice and ocean.

Fig. 11: Have you performed a similar significance test as in Fig. 9? Maybe the plot is getting quite busy with all the symbols then, but mentioning in the text whether these differences are significant or not would be helpful.
We did a similar significance test now for Fig. 11 (Fig. 12 in revised version) and show the results in the figure. Most differences are significant because of the large number of data points.

L492: do you mean larger optical thickness due to a larger vertical extent?
Yes, we added an information that due to the larger vertical extent the optical thickness is also increased.

Technical corrections
L20: I suggest citing a specific chapter (e.g. Chapter 7, Forster and Storelvmo et al. 2021) instead of the entire WGI report.
L22: showed
L36: delete 'differ'
L46: shows
L49: underestimated
L50: showed as well
L51: single-column model simulations
L60: delete 'a'
L79: split into
L108: provides
L111: IFS has been introduced earlier
L180: comma after percentile
L188: separate clouds
L209: delete 'further'
L219: single-layer, as a function single-layer cloud profiles, as a function
L236: also showed
Figure 5: Please increase the size of the figure a bit.
L243+245+250+Fig. 5 caption+...: I would use attenuated instead of extinguished
L263: such as the cloud phase as a function of...
Figure 6 caption: with one vertical profile having a horizontal... (instead of one vertical profiles)
L284: result
L286: ground-based
L303: CTT has been introduced before
L328: temperatures
Fig. 7 caption: the liquid fraction of each profile (not profiles)
Fig. 8: I suggest labelling the x axes 'Fraction of liquid/mixed-phase pixels' to clearly distinguish it from the liquid fraction that has been used up to now.
L357: further research is needed
L361: ground-based
Fig. 9 caption: clouds over sea ice
Fig. 12 caption: left panel shows
Fig. 12: could you increase the size of the figure a bit?
L420: seem to play
L465: further analysis shows
L539: compared to considering all cloud profiles
L540: and not tropical
L553: and 2B-FLXHR-LIDAR for CRE?

L588: in lower parts of the cloud

Thanks for the careful reading. We changed the technical corrections as suggested.

**References**

Grégory V. Cesana, Théodore Khadir, Hélène Chepfer, and Marjolaine Chiriaco. Southern Ocean Solar Reflection Biases in CMIP6 Models Linked to Cloud Phase and Vertical Structure Representations. *Geophysical Research Letters*, 49(22):e2022GL099777, 2022. ISSN 1944-8007. doi: 10.1029/2022GL099777. https://onlinelibrary.wiley.com/doi/abs/10.1029/2022GL099777.

---

## Author Response (AR2)

**Response to reviewer comments on "Characterisation of low-base and mid-base clouds and their thermodynamic phase over the Southern and Arctic Ocean" by B. Dietel, O. Sourdeval, C. Hoose**

We thank the anonymous reviewer for their recommendations and comments on the manuscript. Please find the detailed responses below in blue.

**Reply to reviewer comments:**

I was somewhat disappointed in reading the response to my main comments and the revised manuscript. Only one major revision was done in response to this; the inclusion of a more significant uncertainty section. Unfortunately this section is rather superficial and reads like "OK, we'll include some text on uncertainty to please the reviewer" but nothing else in the study has changed much to alleviate my concerns in this respect.

Thank you for the critical, yet constructive feedback. We have revised our analysis and the manuscript significantly in response to the criticism. In particular, we have now removed all clouds below 500m from the analysis. We have also repeated the analyses with different latitude ranges for the Arctic and Southern Ocean regions and present the results here in the replies.

I still contend that the fact that CloudSat can't be used below 500 m combined with the fact that CALIPSO becomes attenuated for optically thick clouds means that a large part of the low clouds will be misrepresented. In the Arctic, clouds are dominated by low clouds with cloud tops below 1 km and cloud bases below a few hundred meters. Other studies indicate these are often mixed phase; in this case a liquid layer at the cloud top that precipitates ice. In that case the cloud top will be easily picked up by the radar but not the cloud interior and therefore the cloud base can often not be detected. This I believe is also reflected in low clouds being liquid more often than mix-phase which is contrary to many studies showing that liquid-only clouds are relatively few. Another effect is that the liquid fraction decreases with height into the cloud, which is just unphysical. There is nothing one can do; what is not observed is just not observed. Consequently, there are also large differences between the results for when the lidar is attenuated or not. But the results for the attenuated cases is a mix of clouds where the radar could be used (CBH > ∼500 m) and where the radar cannot be used. The proper comparison should therefore be only cases where you know the CBH is higher than 500 m. When I read the text, there is actually a lot of speculation on this topic, but it never floats to the top; this needs to be handled up front. I therefore think that the uncertainty analysis is kind of useless as it doesn't contain any analysis of what the effect of these uncertainties mean for the results in this study. There is just a list of uncertain things; the study then proceeds ignoring most of them.

In the revised version of the manuscript, a threshold for the CBH of 500 m has been applied as suggested. To explain why we have not done this before despite the known problems with the radar ground clutter, a drawback is that we also loose information which would be available from the lidar for very thin low clouds, which is the advantage of the the combined usage of radar and lidar information. The lidar extinguishes within about 300 m (Danker et al., 2022), which means there would still be some information for very thin low clouds from the lidar. But we also see the point that especially for the lowest 500 m the uncertainty of the cloud phase is very high due to the missing information from CloudSat . We therefore applied a filter to the entire analysis and repeated all calculations only considering clouds with a cloud base height larger than 500 m to reduce uncertainties introduced by ground clutter from the CloudSat signal, as suggested by the reviewer.

We replaced the figures and adapted the numbers within the text accordingly. We will not list each line where numbers have changed, but instead list here the main changes in the results and figures. The exact changes of all numbers can be found in the marked-up manuscript (latexdiff) version.

- The largest changes from not considering cloud profiles with CBH ≤ 500 m can be found in Fig.

4 and Fig. 6. Fig. 4 shows the fraction of cloud type profiles, and we can see a strong decrease in the relative frequency of low-level clouds from 25.6% (21.5%) to 15.8% (8.6%) for the Southern Ocean (Arctic Ocean) while other cloud types with low CBH (HML, ML) only show a slight decrease in the relative occurrence. Fig. 6 shows the relative occurrence frequencies of the cloud phases for different cloud types. The strongest difference can be seen in the reduced fraction of liquid low-level clouds from 63% (62%) to 40% (53%) over the Southern Ocean (Arctic Ocean) and a corresponding increase of the relative mixed-phase fraction. Mid-low-level and high-mid-low-level clouds affected by the CBH threshold only show small changes in the results. Regarding the results when only profiles are considered where the lidar is not extinguished, we still see an increase of the relative fraction of liquid clouds, which is expected, as the analysis is then more focused on thin clouds which can be penetrated by the lidar.

- The analysis of the horizontal and the vertical extent of clouds (Fig. 7) mainly shows a decreased horizontal extent of low-level clouds which is expected, as less cloud profiles are considered as low-level clouds, and therefore more gaps between profiles considered as the same cloud type along the satellite track occur reducing the calculated horizontal extent of "connected" cloud profiles.

- Regarding the liquid fraction as a function of CTT in Fig. 8 we see a small decrease of the liquid fraction of low-level clouds compared to the previous analysis, so that the liquid fraction of low-level clouds and mid-level clouds are more similar for some CTTs. But all our main points of the discussion are still valid and have not changed. The minima around -15°C and -5°C show up even more clearly.

- In Fig. 9 the vertical phase distribution is shown. We can see that the increased liquid fraction at cloud base is reduced, but is still clearly visible for low-level clouds. Uncertainties due to attenuation or multiple scattering may play a role, but on the other hand for shallow clouds, it is also not unphysical that the first ice forms at cloud top in liquid clouds, where the temperature is coldest, while only at a second step the ice grows strongly at the expense of the liquid droplets, leading to the sedimentation of ice crystals and the typically know mixed-phase cloud structure with liquid at cloud top and large ice crystals below.

- The analysis of the liquid fraction over open ocean and sea ice (Fig. 10) shows slightly lower absolute values, but still higher liquid fractions in clouds over sea ice than in clouds over open ocean for mid-level and mid-low-level clouds over the Southern Ocean and low-level clouds over both regions.

- Fig. 11 and Fig. 13 show almost no differences compared to the previous analysis.

- Fig. 12 shows as well similar results as before and the strange positive values for low CTTs in low-level clouds over the Arctic Ocean regarding sea salt concentration disappeared.

- The cloud radiative effects in Fig. 14 show similar results with the only difference that liquid low-level clouds over the Southern Ocean show now a higher SWCRE than mixed-phase low-level clouds over the Southern Ocean, while it was lower in the previous analysis. Our assumptions is that the exclusion of very thin low layers with a low SWCRE led to an increased SWCRE.

- The changes of the contribution of the cloud types to the SWCRE and LWCRE (Fig. 15) are in line with the changes of the cloud type frequency shown in Fig. 4 leading to a reduced contribution of low-level clouds, but they still show the largest contribution compared to other cloud types in the SWCRE at TOA.

In summary, we can see that most of our results of this study don't change, but reducing the uncertainties introduced by ground clutter strengthens our findings.

My second main objection on comparing Arctic and Southern Oceans with the definitions used here, did not lead to any significant revision; the authors just state that because models are crappy at both it is relevant to compare. I disagree. First, the Arctic Ocean does not extend to 60N; using that definition you also include a fair proportion of the Atlantic and Pacific Oceans. You also include the

tail ends of the extratropical storm tracks, while for the Southern Ocean you include the entire mid-latitudes. But clouds in polar regions are dominated by other clouds than clouds in the midlatitudes. The midlatitudes are affected by convection, deep and shallow, whereas the largest proportion of the Arctic has almost no convection at all and frontal clouds prevail in the midlatitudes; not so much in the polar regions. Finally, a large part of the Arctic remains ice covered even in summer whereas the Antarctic is essentially ice free in summer; in fact the largest part of the Southern Ocean, as defined here, never has any sea ice ever, not even in winter. These differences affect the boundary layer and especially the low clouds; it also affects affects the aerosols, as sea ice essentially cuts off sea spray, and the CRE, since surface albedo is a factor. In summary, this makes the comparison almost impossible interpret. Apples against oranges as it were...

We fully agree that the meteorology and the surface conditions are different between the Northern and Southern high latitudes, but the cloud physics appear to be very similar despite these differences. Therefore, we uphold that not only regions with the exact same conditions can be compared, but that universal relationships can be found from their manifestation in different cloud regimes. In addition, while many studies investigate cloud phase by averaging over all clouds all over the world without further distinction, we categorize cloud types based on their base and top height, which for example can be seen as an indirect way of distinguishing between deep convection and more shallow clouds. Finally, many of the mentioned differences between the Northern and Southern regions (like sea ice coverage or aerosols) are directly analyzed as part of the study within subchapters (see Chap.5.2.4).

Specifically, both of the regions contain:

- areas covered by sea ice,
- areas with open ocean with varying SST,
- cloud profiles with low and high aerosol content,
- areas where convection can occur, and areas where more stratiform clouds can occur,
- and areas of extratropical cyclones.

Contrary to point observations or case studies, we analyze highly resolved vertical cloud profiles over rather large regions using data of two full years, leading to a large dataset covering various conditions withing a parameter space spanned by various variables. This enables a statistical analysis including the investigation of varying conditions like sea ice or aerosols.

We partly agree with the criticism regarding the denomination of the analyzed regions. The reviewer is correct that the "Arctic Ocean" is usually defined as a nearly landlocked ocean consisting of a deep central basin surrounded by seven epicontinental seas, i.e. the Barents, Kara, Laptev, East Siberian, Chukchi, Beaufort, and Lincoln Seas (Jakobsson et al., 2004) and does not extend to 60°N. Nevertheless, the International Hydrographic Organization also includes the Greenland, Norwegian, Iceland, and White Seas; Baffin and Hudson Bays; Davis and Hudson Straits; and the waterways of the Canadian Arctic Archipelago (Jakobsson et al., 2004). We have explained the definition more clearly in the manuscript and now used the term "Arctic marine regions". Meanwhile, we have not changed our definition of the Southern Ocean because it is in agreement with the majority of related literature, as shown below. We changed our title from "[...] Southern and Arctic Ocean" to "[...] Southern Ocean and Arctic marine regions" to clarify that we investigate all seas over the Arctic region including parts of the northern Atlantic and northern Pacific besides the Arctic Ocean. The "Arctic region" is often defined as a region north of 60 °as e.g. in Cesana et al. (2023); Wendisch et al. (2023); Lawrence et al. (2019); Schacht et al. (2019); Comiso (2003). Table 1 lists the analyzed latitudes of several studies investigating cloud over the Southern Ocean, which mostly begin at latitudes of 40°S.
To investigate whether the selection of the latitude range has an impact on the results, we have repeated key parts of the analysis for different latitude ranges (60-82°N and 40-82°S as previously chosen versus 60-70 and 70-80 degrees in both hemispheres). The results of these analyses are included in the following. The cloud phase distribution (Fig. R1) only varies slightly for different latitude bands. The vertical liquid fraction shows very similar results regarding different latitude bands over the Arctic, but shows differences over the Southern Ocean. Low-level, mid-level, and mid-low-level

Table 1: Definitions of the Southern Ocean in previous literature investigating clouds over the Southern Ocean.

| Analysed region | Literature | Title |
| --- | --- | --- |
| 40°S - 60°S | Wall et al. (2022) | Observational Constraints on Southern Ocean Cloud-Phase Feedback |
| 40°S - 72°S | Bodas-Salcedo et al. (2016) | Large Contribution of Supercooled Liquid Clouds to the Solar Radiation Budget of the Southern Ocean |
| 40°S - 70°S | Cesana et al. (2023) | The correlation between Arctic sea ice, cloud phase and radiation using A-train satellites |
| 40°S - 60°S | Coopman et al. (2021) | Analyzing the Thermodynamic Phase Partitioning of Mixed Phase Clouds Over the Southern Ocean Using Passive Satellite Observations |
| 30°S - 75°S | D'Alessandro et al. (2019) | Cloud Phase and Relative Humidity Distributions over the Southern Ocean in Austral Summer Based on In Situ Observations and CAM5 Simulations |
| 50°S - 80°S | D'Alessandro et al. (2021) | Characterizing the Occurrence and Spatial Heterogeneity of Liquid, Ice, and Mixed Phase Low-Level Clouds Over the Southern Ocean Using in Situ Observations Acquired During SOCRATES |
| 40°S - 65°S | Danker et al. (2021) | Exploring Relations between Cloud Morphology, Cloud Phase, and Cloud Radiative Properties in Southern Ocean Stratocumulus Clouds |
| 40°S - 65°S | Huang et al. (2012) | A study on the low-altitude clouds over the Southern Ocean using the DARDAR-MASK |
| 40°S - 60°S | Huang et al. (2015) | A-Train Observations of Maritime Midlatitude Storm-Track Cloud Systems: Comparing the Southern Ocean against the North Atlantic |
| 40°S - 70°S | Kay et al. (2016) | Global Climate Impacts of Fixing the Southern Ocean Shortwave Radiation Bias in the Community Earth System Model (CESM) |

[Figure]

Figure R1: Cloud phase distributions of various cloud types (rows) over the Arctic ocean and the Southern Ocean. The outer pie charts include all cloud profiles of a cloud type, while the inner pie charts only consider cloud profiles, where the lidar signal is not fully attenuated by the cloud. The left panel shows the results for the definitions used in our manuscript. The middle panel shows the results only considering latitudes between 60°N/S and 70°N/S, while the right panel shows results for latitudes between 70°N/S and 80°N/S. All analyses exclude cloud profiles over land surfaces.

clouds show higher liquid fractions for higher latitudes regarding the same cloud top temperature. Nevertheless, Fig. R3 shows that the comparison between different cloud types over both regions looks very similar for different latitudes.

We also analyzed further aspects of different conditions influencing our results like sea ice cover on CRE (shown in Fig. R4). As expected we can see a reduced SWCRE over sea ice, but the relative behavior of the CRE does not change if we only consider clouds over open ocean (panel c,d) compared to the discussed results in the manuscripts, where cloud profiles over sea ice are included (panel a, b).

[Figure]

Figure R2: Comparison of the vertical liquid fraction as function of cloud top temperature (CTT) considering different latitude bands of our analyzed regions. The upper row refers northern latitudes, while the bottom row refer to southern latitudes.

[Figure]

Figure R3: Comparison of the manuscript results regarding the liquid fraction considering different latitude bands.

[Figure]

Figure R4: TOACRE for different cloud types. Panel (a) and (b) show the results shown in the paper considering profiles over ocean and sea ice, while panel (c) and (d) distinguish between open ocean and sea ice. Sea ice defined by a sea ice concentration equal or larger than 80%.

In summary, I must continue to insist on a major revision. I have numerous minor comments that I'm saving until I can review a manuscript that takes my objections above seriously.

We believe that our additional analyses and revisions address the major points listed above and are looking forward to the additional minor comments.

**References**

A. Bodas-Salcedo, P. G. Hill, K. Furtado, K. D. Williams, P. R. Field, J. C. Manners, P. Hyder, and S. Kato. Large Contribution of Supercooled Liquid Clouds to the Solar Radiation Budget of the Southern Ocean. *Journal of Climate*, 29(11):4213–4228, June 2016. ISSN 0894-8755, 1520-0442. doi: 10.1175/JCLI-D-15-0564.1. https://journals.ametsoc.org/view/journals/clim/29/11/jcli-d-15-0564.1.xml.

Grégory V. Cesana, Olivia Pierpaoli, Matteo Ottaviani, Linh Vu, and Zhonghai Jin. The correlation between Arctic sea ice, cloud phase and radiation using A-train satellites. *EGUsphere*, pages 1–17, December 2023. doi: 10.5194/egusphere-2023-2940. https://egusphere.copernicus.org/preprints/2023/egusphere-2023-2940/.

Josefino C. Comiso. Warming Trends in the Arctic from Clear Sky Satellite Observations. *Journal of Climate*, 16(21):3498–3510, November 2003. ISSN 0894-8755, 1520-0442. doi: 10.1175/1520-0442(2003)016⟨3498:WTITAF⟩2.0.CO;2. https://journals.ametsoc.org/view/journals/clim/16/21/1520-0442_2003_016_3498_wtitaf_2.0.co_2.xml.

Q. Coopman, C. Hoose, and M. Stengel. Analyzing the Thermodynamic Phase Partitioning of Mixed Phase Clouds Over the Southern Ocean Using Passive Satellite Observations. *Geophysical Research Letters*, 48(7):e2021GL093225, 2021. ISSN 1944-8007. doi: 10.1029/2021GL093225. https://onlinelibrary.wiley.com/doi/abs/10.1029/2021GL093225.

John J. D'Alessandro, Minghui Diao, Chenglai Wu, Xiaohong Liu, Jorgen B. Jensen, and Britton B. Stephens. Cloud Phase and Relative Humidity Distributions over the Southern Ocean in Austral Summer Based on In Situ Observations and CAM5 Simulations. *Journal of Climate*, 32(10):2781–2805, May 2019. ISSN 0894-8755, 1520-0442. doi: 10.1175/JCLI-D-18-0232.1. https://journals.ametsoc.org/view/journals/clim/32/10/jcli-d-18-0232.1.xml.

John J. D'Alessandro, Greg M. McFarquhar, Wei Wu, Jeff L. Stith, Jorgen B. Jensen, and Robert M. Rauber. Characterizing the Occurrence and Spatial Heterogeneity of Liquid, Ice, and Mixed Phase Low-Level Clouds Over the Southern Ocean Using in Situ Observations Acquired During SOCRATES. *Journal of Geophysical Research: Atmospheres*, 126(11):e2020JD034482, 2021. ISSN 2169-8996. doi: 10.1029/2020JD034482. https://agupubs.onlinelibrary.wiley.com/doi/abs/10.1029/2020JD034482.

Jessica Danker, Odran Sourdeval, Isabel L. McCoy, Robert Wood, and Anna Possner. Exploring Relations between Cloud Morphology, Cloud Phase, and Cloud Radiative Properties in Southern Ocean Stratocumulus Clouds. *Atmospheric Chemistry and Physics Discussions*, pages 1–26, November 2021. ISSN 1680-7316. doi: 10.5194/acp-2021-926. https://acp.copernicus.org/preprints/acp-2021-926/.

Jessica Danker, Odran Sourdeval, Isabel L. McCoy, Robert Wood, and Anna Possner. Exploring relations between cloud morphology, cloud phase, and cloud radiative properties in Southern Ocean's stratocumulus clouds. *Atmospheric Chemistry and Physics*, 22(15):10247–10265, August 2022. ISSN 1680-7316. doi: 10.5194/acp-22-10247-2022. https://acp.copernicus.org/articles/22/10247/2022/.

Yi Huang, Steven T. Siems, Michael J. Manton, Alain Protat, and Julien Delanoë. A study on the low-altitude clouds over the Southern Ocean using the DARDAR-MASK. *Journal of Geophysical Research: Atmospheres*, 117(D18), 2012. ISSN 2156-2202. doi: 10.1029/2012JD017800. https://agupubs.onlinelibrary.wiley.com/doi/abs/10.1029/2012JD017800.

Yi Huang, Alain Protat, Steven T. Siems, and Michael J. Manton. A-Train Observations of Maritime Midlatitude Storm-Track Cloud Systems: Comparing the Southern Ocean against the North Atlantic. *Journal of Climate*, 28(5):1920–1939, March 2015. ISSN 0894-8755. doi: 10.1175/JCLI-D-14-00169.1. https://journals.ametsoc.org/jcli/article/28/5/1920/35207/A-Train-Observations-of-Maritime-Midlatitude-Storm.

M. Jakobsson, A. Grantz, Y. Kristoffersen, R. Macnab, R. W. MacDonald, E. Sakshaug, R. Stein, and W. Jokat. The Arctic Ocean: Boundary Conditions and Background Information. In Ruediger Stein and Robie W. MacDonald, editors, *The Organic Carbon Cycle in the Arctic Ocean*, pages 1–32. Springer, Berlin, Heidelberg, 2004. ISBN 978-3-642-18912-8. doi: 10.1007/978-3-642-18912-8_1. https://doi.org/10.1007/978-3-642-18912-8_1.

Jennifer E. Kay, Casey Wall, Vineel Yettella, Brian Medeiros, Cecile Hannay, Peter Caldwell, and Cecilia Bitz. Global Climate Impacts of Fixing the Southern Ocean Shortwave Radiation Bias in the Community Earth System Model (CESM). *Journal of Climate*, 29(12):4617–4636, June 2016. ISSN 0894-8755, 1520-0442. doi: 10.1175/JCLI-D-15-0358.1. https://journals.ametsoc.org/view/journals/clim/29/12/jcli-d-15-0358.1.xml.

Heather Lawrence, Jacky Farnan, Niels Bormann, and Peter Bauer. An Assessment of the use of observations in the Arctic at ECMWF. April 2019. https://www.ecmwf.int/en/elibrary/80977-assessment-use-observations-arctic-ecmwf.

Jacob Schacht, Bernd Heinold, Johannes Quaas, John Backman, Ribu Cherian, Andre Ehrlich, Andreas Herber, Wan Ting Katty Huang, Yutaka Kondo, Andreas Massling, P. R. Sinha, Bernadett Weinzierl, Marco Zanatta, and Ina Tegen. The importance of the representation of air pollution emissions for the modeled distribution and radiative effects of black carbon in the Arctic. *Atmospheric Chemistry and Physics*, 19(17):11159–11183, September 2019. ISSN 1680-7316. doi: 10.5194/acp-19-11159-2019. https://acp.copernicus.org/articles/19/11159/2019/.

Casey J. Wall, Trude Storelvmo, Joel R. Norris, and Ivy Tan. Observational Constraints on Southern Ocean Cloud-Phase Feedback. *Journal of Climate*, 35(15):5087–5102, August 2022. ISSN 0894-8755, 1520-0442. doi: 10.1175/JCLI-D-21-0812.1. https://journals.ametsoc.org/view/journals/clim/35/15/JCLI-D-21-0812.1.xml.

M. Wendisch, M. Brückner, S. Crewell, A. Ehrlich, J. Notholt, C. Lüpkes, A. Macke, J. P. Burrows, A. Rinke, J. Quaas, M. Maturilli, V. Schemann, M. D. Shupe, E. F. Akansu, C. Barrientos-Velasco, K. Bärfuss, A.-M. Blechschmidt, K. Block, I. Bougoudis, H. Bozem, C. Böckmann, A. Bracher, H. Bresson, L. Bretschneider, M. Buschmann, D. G. Chechin, J. Chylik, S. Dahlke, H. Deneke, K. Dethloff, T. Donth, W. Dorn, R. Dupuy, K. Ebell, U. Egerer, R. Engelmann, O. Eppers, R. Gerdes, R. Gierens, I. V. Gorodetskaya, M. Gottschalk, H. Griesche, V. M. Gryanik, D. Handorf, B. Harm-Altstädter, J. Hartmann, M. Hartmann, B. Heinold, A. Herber, H. Herrmann, G. Heygster, I. Höschel, Z. Hofmann, J. Hölemann, A. Hünerbein, S. Jafariserajehlou, E. Jäkel, C. Jacobi, M. Janout, F. Jansen, O. Jourdan, Z. Jurányi, H. Kalesse-Los, T. Kanzow, R. Käthner, L. L. Kliesch, M. Klingebiel, E. M. Knudsen, T. Kovács, W. Körtke, D. Krampe, J. Kretzschmar, D. Kreyling, B. Kulla, D. Kunkel, A. Lampert, M. Lauer, L. Lelli, A. von Lerber, O. Linke, U. Löhnert, M. Lonardi, S. N. Losa, M. Losch, M. Maahn, M. Mech, L. Mei, S. Mertes, E. Metzner, D. Mewes, J. Michaelis, G. Mioche, M. Moser, K. Nakoudi, R. Neggers, R. Neuber, T. Nomokonova, J. Oelker, I. Papakonstantinou-Presvelou, F. Pätzold, V. Pefanis, C. Pohl, M. van Pinxteren, A. Radovan, M. Rhein, M. Rex, A. Richter, N. Risse, C. Ritter, P. Rostosky, V. V. Rozanov, E. Ruiz Donoso, P. Saavedra Garfias, M. Salzmann, J. Schacht, M. Schäfer, J. Schneider, N. Schnierstein, P. Seifert, S. Seo, H. Siebert, M. A. Soppa, G. Spreen, I. S. Stachlewska, J. Stapf, F. Stratmann, I. Tegen, C. Viceto, C. Voigt, M. Vountas, A. Walbröl, M. Walter, B. Wehner, H. Wex, S. Willmes, M. Zanatta, and S. Zeppenfeld. Atmospheric and Surface Processes, and Feedback Mechanisms Determining Arctic Amplification: A Review of First Results and Prospects of the (AC)3 Project. *Bulletin of the American Meteorological Society*, 104 (1):E208–E242, January 2023. ISSN 0003-0007, 1520-0477. doi: 10.1175/BAMS-D-21-0218.1. https://journals.ametsoc.org/view/journals/bams/104/1/BAMS-D-21-0218.1.xml.

---

## Author Response (AR3)

Dear Matthias,

Thank you for the positive comments on our manuscript. We have changed "Arctic Ocean" to "Arctic marine regions" throughout the manuscript, including the figures. Furthermore, we have revised figures 5, 8, 9, 10, A01 and A02 to make them readable for people with colour vision deficiencies.

Best regards
Corinna